# Optimizing Estimators of Squared Calibration Errors in Classification

**Sebastian G. Gruber**                                           *sebastian.gruber@dkfz.de*
*German Cancer Consortium (DKTK), partner site Frankfurt/Mainz,*
*a partnership between DKFZ and UCT Frankfurt-Marburg, Germany, Frankfurt am Main, Germany*
*German Cancer Research Center (DKFZ), Heidelberg, Germany*
*Goethe University Frankfurt, Germany*

**Francis Bach**
*Inria, Ecole Normale Supérieure, PSL Research University, Paris, France*

**Reviewed on OpenReview:** *https://openreview.net/forum?id=BPDVZajOW5*

## Abstract

In this work, we propose a mean-squared error-based risk that enables the comparison and optimization of estimators of squared calibration errors in practical settings. Improving the calibration of classifiers is crucial for enhancing the trustworthiness and interpretability of machine learning models, especially in sensitive decision-making scenarios. Although various calibration (error) estimators exist in the current literature, there is a lack of guidance on selecting the appropriate estimator and tuning its hyperparameters. By leveraging the bilinear structure of squared calibration errors, we reformulate calibration estimation as a regression problem with independent and identically distributed (i.i.d.) input pairs. This reformulation allows us to quantify the performance of different estimators even for the most challenging calibration criterion, known as canonical calibration. Our approach advocates for a training-validation-testing pipeline when estimating a calibration error on an evaluation dataset. We demonstrate the effectiveness of our pipeline by optimizing existing calibration estimators and comparing them with novel kernel ridge regression-based estimators on real-world image classification tasks.

## 1 Introduction

In the field of machine learning, classification tasks involve predicting discrete class labels for given instances (Bishop & Nasrabadi, 2006). As these models are increasingly being employed in critical applications such as healthcare (Haggenmüller et al., 2021), autonomous driving (Feng et al., 2020), weather forecasting (Gneiting & Raftery, 2005), and financial decision-making (Frydman et al., 1985), the need for reliable and interpretable predictions has become of critical importance. A key aspect of reliability in classification models is the calibration of their predicted probabilities (Murphy & Winkler, 1977; Hekler et al., 2023). Calibration refers to the alignment between predicted probabilities and the true likelihoods of outcomes, ensuring that predictions are not only accurate but also meaningful in terms of their confidence scores (Murphy, 1973). Despite the advancements in model architectures and learning algorithms, many modern classifiers, such as deep neural networks, are prone to producing overconfident predictions (Minderer et al., 2021). This overconfidence can be attributed to several factors, including the model's complexity, training data limitations, and inherent biases in learning processes (Guo et al., 2017). Consequently, even models that achieve high accuracy might suffer from poor calibration, leading to potential misinterpretations and suboptimal decisions.

To quantify the extent to which a model is miscalibrated, calibration errors have been introduced (Naeini et al., 2015). However, their estimators are usually biased (Roelofs et al., 2022) and inconsistent (Vaicenavicius et al., 2019). Other calibration errors with unbiased estimators exist but they lack theoretical derivation and

are difficult to interpret (Widmann et al., 2019; 2021; Marx et al., 2024). This, in turn, is highly problematic since we cannot quantify how reliable a model is if we either do not know how reliable the metric is or how to interpret it. In this work, we tackle the former problem. We propose mean-squared error risk minimization to find an optimal calibration estimator for any squared calibration error. This risk can be applied in any practical scenario to compare and select different estimators, and works for all notions of calibration, even for the notoriously difficult canonical calibration. Our **contributions** are as follows:

- We propose a novel risk applicable for squared calibration estimators in Section 3.1, which represents an optimization objective for comparing calibration estimators proposed by the literature.

- We formulate a calibration-evaluation pipeline based on our risk in Section 3.2.2, which allows to optimize calibration estimators used by the literature.

- We propose novel kernel ridge regression-based calibration estimators in Section 4, and compare these with optimized baselines on common image classification models in Section 5.

## 2 Background

In the following, we offer an extensive introduction to the background of this work. First, we offer a more extensive motivation for calibration. Second, we give briefly measure theoretic preliminaries, followed by the different notions of calibration used for classification by the literature. We then introduce and discuss commonly used estimators for these notions.

### 2.1 An Exemplary Motivation for Calibration

Uncertainty calibration in classification is directly connected to practitioners' concerns in real-world application as the following example demonstrates. Assume we are in a sensitive classification task, e.g., we aim to predict the classes $Y \in \{$*No-Tumor*, *Benign-Tumor*, *Malignant-Tumor*$\}$ for a given patient $X$ via a classifier $f$. In such a setting, the practitioner may be interested in the uncertainty of a given prediction $f(X)$ to assess how reliable the prediction is. One possible uncertainty is the predicted probability for each outcome, e.g., $f(X) = \{$*No-Tumor*: 0.7, *Benign-Tumor*: 0.2, *Malignant-Tumor*: 0.1 $\}$, which aims to predict the unknown ground truth probabilities $\mathbb{P}(Y = $*No-Tumor*$ \mid X), \mathbb{P}(Y = $*Benign-Tumor*$ \mid X)$, and $\mathbb{P}(Y = $*Malignant-Tumor*$ \mid X)$. However, similar to how the predicted class may be untruthful, so may the predicted probabilities. To see this, assume we also have a dataset of 10.000 prediction-target pairs in the previous example. Now, further assume there exist 100 instances of prediction-target pairs within this dataset with the above prediction. Ideally, if the predicted probabilities were truthful, then among these 100 instances, there would be 70 instances of *No-Tumor*, 20 instances of *Benign-Tumor*, and 10 instances of *Malignant-Tumor*. If this holds for all predictions, we refer to the underlying classifier as being calibrated. In summary, the central question for calibration is, given a probability prediction $f(X)$, how closely do the outcome probabilities $\mathbb{P}(Y \mid f(X))$ match $f(X)$? Squared calibration errors are then used to quantify the expected square difference between prediction and outcome probabilities (Murphy, 1973). In the following, we formulate these concepts in a mathematically rigorous manner similar to (Vaicenavicius et al., 2019).

### 2.2 Measure Theoretic Preliminaries

Throughout this work we use measure theoretic definitions to formalize our contribution, its assumptions, and its limitations. Specifically, we assume a measure space $(\Omega, \mathscr{F}, \mu)$ is given. We associate with a random variable $X: \Omega \to \mathscr{X}$ (a measureable function) a probability space $(\mathscr{X}, \mathscr{F}_X, \mathbb{P}_X)$ with $\sigma$-field $\mathscr{F}_X = \{X(A) \mid A \in \mathscr{F}\}$ and pushforward measure $\mathbb{P}_X = \mu \circ X^{-1}$. Further, we may also assume a second random variable $Y: \Omega \to \mathscr{Y}$ such that we have a joint probability space $(\mathscr{X} \times \mathscr{Y}, \mathscr{F}_{XY}, \mathbb{P}_{XY})$, where $\mathscr{F}_{XY}$ and $\mathbb{P}_{XY}$ are defined analogous as $\mathscr{F}_X$ and $\mathbb{P}_X$. Further, if a random variable $Y$ is discrete, we use $\mathbb{P}_Y$ to represent both its probability measure and the associated probability vector in the simplex $\Delta^d := \left\{(p_1, \ldots, p_k)^\top \in [0,1]^d \mid \sum_{i=1}^d p_i = 1\right\}$, where $d$ is the number of unique outcomes. In the same sense we may use $\mathbb{P}_{Y|X} := \frac{d\mathbb{P}_{XY}}{d\mathbb{P}_X}$ as a measurable function $\mathscr{X} \to \Delta^d$.

Finally, we say a statement holds $\mu$-almost surely ($\mu$-a.s.) if it is true except on a set of $\mu$-measure zero (Capiński & Kopp, 2004).

### 2.3 Mean-Squared Error Risk Minimization and Calibration

The mean-squared error (MSE) is one of the most common loss functions used in regression problems, see, e.g., (Efron, 1994; Schölkopf & Smola, 2002; Bishop & Nasrabadi, 2006; Goodfellow et al., 2016; Murphy, 2022; Bach, 2024). Its expected loss for a vector-valued sample $Y \sim \mathbb{P}_Y$ from a target distribution $\mathbb{P}_Y$ and a prediction $c \in \mathbb{R}^d$ is defined by

$$L_{\mathrm{MSE}}(c, \mathbb{P}_Y) \coloneqq \mathbb{E}_{Y \sim \mathbb{P}_Y}\left[\|c - Y\|^2\right]. \tag{1}$$

It holds that the expectation of the target term in the difference is the unique minimizer, i.e.,

$$\mathbb{E}[Y] = \underset{c \in \mathbb{R}^d}{\arg\min}\, L_{\mathrm{MSE}}(c, \mathbb{P}_Y). \tag{2}$$

Now, let's introduce another random variable $X$ such that $(X, Y) \sim \mathbb{P}_{XY}$ follows a joint distribution and $X$ can be used as input for a regression model $m \colon \mathscr{X} \to \mathbb{R}^d$ to approximate the target conditional distribution $m^*(x) \coloneqq \mathbb{E}[Y \mid X = x]$. Then, the corresponding risk of $m$ is defined by

$$\mathscr{R}_{\mathrm{MSE}}(m) \coloneqq \mathbb{E}_{X \sim \mathbb{P}_X}\left[L_{\mathrm{MSE}}(m(X), \mathbb{P}_{Y|X})\right] = \mathbb{E}_{(X,Y) \sim \mathbb{P}_{XY}}\left[\|m(X) - Y\|^2\right]. \tag{3}$$

Given $m \neq m^*$, where the inequality holds for a set of positive probability mass, it follows that

$$\mathscr{R}_{\mathrm{MSE}}(m) > \mathscr{R}_{\mathrm{MSE}}(m^*). \tag{4}$$

However, we have the measure theoretic limitation that there might be a null set $A \in \mathscr{F}_X$ and some $\tilde{m}$ such that $\tilde{m}(x) \neq m^*(x)$ for all $x \in A$ while $\mathscr{R}_{\mathrm{MSE}}(\tilde{m}) = \mathscr{R}_{\mathrm{MSE}}(m^*)$. This limitation of risk minimization is usually irrelevant in machine learning applications. However, in our case, it will require additional theoretical assumptions on the target conditional distribution to make risk minimization a usable tool for assessing calibration estimators.

The MSE can also be used for classification tasks with a one-hot encoded target in Equation 1, often referred to as Brier score (Brier, 1950). Then, Murphy (1973) introduces the concept of calibration for a classifier $f \colon \mathscr{X} \to \Delta^d$ by showing that

$$\mathscr{R}_{\mathrm{MSE}}(f) = \mathbb{E}_{X \sim \mathbb{P}_X}\left[\|f(X) - \mathbb{P}_{Y|f(X)}\|^2\right] - \mathbb{E}_{X \sim \mathbb{P}_X}\left[\|\mathbb{P}_Y - \mathbb{P}_{Y|f(X)}\|^2\right] + \|\mathbb{P}_Y\|^2. \tag{5}$$

The first term on the right-hand side is usually referred to as the calibration term, and the second as sharpness term, coining Equation (5) as calibration-sharpness decomposition of the Brier score (Gneiting et al., 2007; Gruber & Buettner, 2022; Kuleshov & Deshpande, 2022; Gruber et al., 2024; Sun et al., 2024). In the calibration term, $f(X)$ is compared with the target distribution $\mathbb{P}(Y \mid f(X))$ given the full predicted vector. In current literature, this notion of calibration is referred to as canonical calibration (Vaicenavicius et al., 2019; Popordanoska et al., 2022b; Gupta & Ramdas, 2022; Gruber et al., 2024). Formally, we say the model $f$ is **canonically calibrated** if and only if

$$\mathbb{P}(Y = i \mid f(X) = p) = p_i \text{ for all } p \in \Delta^d, i = 1 \dots d. \tag{6}$$

The corresponding $L^2$ **canonical calibration error** is defined as

$$\mathrm{CCE}_2(f) \coloneqq \sqrt{\mathbb{E}_X\left[\|f(X) - \mathbb{P}_{Y|f(X)}\|^2\right]}, \tag{7}$$

which is equal to the calibration term in Equation (5). It holds that $\mathrm{CCE}_2(f) = 0$ if and only if $f$ is canonically calibrated $\mathbb{P}_X$-almost surely.

However, canonical calibration errors are notoriously difficult to estimate and represent a calibration strictness which may not be necessary in practice (Vaicenavicius et al., 2019). Consequently, other notions of calibration have been proposed, which we discuss next.

## 2.4 Alternative Notions of Calibration

Besides canonical calibration, multiple notions of calibration have been introduced in the literature (Zadrozny & Elkan, 2002; Vaicenavicius et al., 2019; Kull et al., 2019; Gupta & Ramdas, 2022). Respective calibration errors assess the degree to which a classifier violates a given notion. In recent literature, the most common notion is top-label confidence calibration (Naeini et al., 2015; Guo et al., 2017; Joo et al., 2020; Kristiadi et al., 2020; Rahimi et al., 2020; Tomani et al., 2021; Minderer et al., 2021; Tian et al., 2021; Islam et al., 2021; Menon et al., 2021; Morales-Álvarez et al., 2021; Gupta et al., 2021; Wang et al., 2021; Fan et al., 2022; Dehghani et al., 2023; Chang et al., 2024). Here, we compare if the predicted top-label confidence $\max_{i \in \mathcal{Y}} f_i(X)$ matches the conditional accuracy $\mathbb{P}(Y = \arg\max_{i \in \mathcal{Y}} f_i(X) \mid \max_{i \in \mathcal{Y}} f_i(X))$ given the prediction. Formally, we say the classifier $f$ is **top-label confidence calibrated** if and only if

$$\mathbb{P}\left(Y = \arg\max_i f_i(X) \,\middle|\, \max_i f_i(X) = p\right) = p \text{ for all } p \in [0, 1]. \tag{8}$$

The corresponding $L^2$ **top-label confidence calibration error** is defined as

$$\mathrm{TCE}_2(f) := \sqrt{\mathbb{E}_X\left[\left(\max_i f_i(X) - \mathbb{P}\left(Y = \arg\max_i f_i(X) \mid \max_i f_i(X)\right)\right)^2\right]}. \tag{9}$$

It holds that $\mathrm{TCE}_2(f) = 0$ if and only if $f$ is top-label confidence calibrated $\mathbb{P}_X$-almost surely. It is easier to estimate than $\mathrm{CCE}_2$ since the target conditional distribution is only based on a scalar random variable independent of the number of classes. However, top-label confidence calibration is a weaker condition than canonical calibration since the implication $\mathrm{CCE}_2(f) = 0 \implies \mathrm{TCE}_2(f) = 0$ does not generally hold in the reverse direction (Gruber & Buettner, 2022).

Other notions of calibration exist, which also reduce the prediction to a scalar, such as class-wise calibration (Zadrozny & Elkan, 2002; Kull et al., 2019; Kumar et al., 2019). Gupta & Ramdas (2022) introduce further of such notions. In general, these notions are transformations of the full probability vectors to a lower dimensional space. Similar to top-label confidence calibration, this makes them easier to estimate than canonical calibration but also turns them into a weaker condition due to the information loss of the transformation (Vaicenavicius et al., 2019; Gruber & Buettner, 2022).

## 2.5 Calibration Estimators

We assume a dataset of i.i.d. samples $(X_1, Y_1), \ldots, (X_n, Y_n) \sim \mathbb{P}_{XY}$ to estimate the calibration of a given classifier $f \colon \mathcal{X} \to \Delta^d$. The most common approach to estimate calibration errors based on scalar conditionals are binning schemes (Naeini et al., 2015; Guo et al., 2017; Minderer et al., 2021; Detlefsen et al., 2022). A prominent estimator is the so-called *expected calibration error* (ECE), which is a binning-based estimator of the $L^1$ top-label confidence calibration error (Guo et al., 2017). In essence, the conditional target distribution $\mathbb{P}(Y = \arg\max_i f_i(X) \mid \max_i f_i(X))$ is estimated via a histogram binning scheme, which places all top-label confidence predictions into mutually distinct bins $B_m := \{i \mid \arg\max_j f_j(X_i) \in I_m\}$, $m = 1, \ldots, M$, based on a partition $\bigcup_m I_m = [0, 1]$. The analogous $L^2$ estimator is given by

$$\mathrm{TCE}_2^{\mathrm{bin}}(f) := \sqrt{\sum_{m=1}^M \frac{|B_m|}{n}\left(\mathrm{acc}(B_m) - \mathrm{conf}(B_m)\right)^2} \tag{10}$$

with $\mathrm{acc}(B) = \frac{1}{|B|}\sum_{i \in B}\mathbf{1}_{Y_i = \arg\max_j f_j(X_i)}$ and $\mathrm{conf}(B) = \frac{1}{|B|}\sum_{i \in B}\arg\max_j f_j(X_i)$ (Kumar et al., 2019). This estimator is primarily suitable for target distributions conditioned on a scalar random variable. The choice of bin intervals $I_1, \ldots, I_M$ is user-defined and the estimator only converges to $\mathrm{TCE}_2$ for an adaptive scheme (Vaicenavicius et al., 2019). Patel et al. (2021) and Roelofs et al. (2022) propose approaches to automatically select appropriate bins. However, in practice, it remains an open challenge to definitively select the optimal choice for a specific dataset and classifier. Analogous binning-based estimators for class-wise calibration exist

in the literature and share these limitations (Kumar et al., 2019; Nixon et al., 2019; Vaicenavicius et al., 2019).

Estimating canonical calibration is more difficult than other notions of calibration due to the target distribution $\mathbb{P}(Y \mid f(X))$ being conditioned on a vector-valued random variable. Popordanoska et al. (2022b) propose to use a kernel density ratio estimator, which is closely related to the Nadaraya-Watson-estimator (Bierens, 1996). The estimator for $\mathrm{CCE}_2$ is given by

$$\mathrm{CCE}_2^{\mathrm{kde}}(f) := \sqrt{\frac{1}{n} \sum_{i=1}^{n} \left\| f(X_i) - \frac{\sum_j k_{\mathrm{dir}}(f(X_j); f(X_i)) e_{Y_j}}{\sum_j k_{\mathrm{dir}}(f(X_j); f(X_i))} \right\|^2}, \tag{11}$$

where $e_i$ refers to the unit vector with a 1 at index $i$ and $k_{\mathrm{dir}}$ is chosen to be the Dirichlet kernel, which is specifically suited for the simplex space (Ouimet & Tolosana-Delgado, 2022). The authors also propose analogous kernel density based estimators for top-label and class-wise calibration errors, which we will denote as $\mathrm{TCE}_2^{\mathrm{kde}}$ and $\mathrm{CWCE}_2^{\mathrm{kde}}$. Even though the kernel density approach is advantageous compared to the binning approach for higher dimensions, it still shares some of its limitations. Popordanoska et al. (2022b) show that the estimator converges in the infinite data limit, but it is still not clear what is the optimal choice of kernel and kernel hyperparameters in the finite data regime. Further, the runtime complexity $\mathrm{CCE}_2^{\mathrm{kde}}$ is in $O(n^2)$.

To summarize, a lot of different approaches have been proposed by the literature to estimate calibration errors. However, it is not clear how to compare different estimators and pinpoint an optimal choice in a finite data setup in practice.

## 3 A Mean-Squared Error Risk for Calibration Estimators

In this section, we present our main contribution: A mean-squared error based risk, which can be applied to compare different calibration estimators in a practical, finite data setup. We first discuss its measure theoretic foundations in Section 3.1, and, then, propose a training-inference pipeline for estimating the calibration error in practice in Section 3.2. This pipeline is analogous to how model training, model selection, and test error evaluation is done in practice in machine learning (Bishop & Nasrabadi, 2006). All formulations are with respect to canonical calibration, since this is the most general case. Other notions, like top-label confidence calibration, can be derived by restricting the canonical case to binary classification. All missing proofs are presented in Appendix C.

### 3.1 Theoretical Definition and Properties

Note that for the $\mathrm{CCE}_2$ calibration error it is sufficient to find a function $h^*: \Delta^d \times \Delta^d \to \mathbb{R}$ such that

$$h^*(p, p') = \langle p - \mathbb{P}_{Y|f(X)=p}, p' - \mathbb{P}_{Y|f(X)=p'} \rangle, \tag{12}$$

since from this follows that

$$\sqrt{\mathbb{E}_X[h^*(f(X), f(X))]} = \mathrm{CCE}_2(f). \tag{13}$$

Indeed, in a later section, we will discover that current estimators already implicitly use such a form. In general, we refer to a function $h: \Delta^d \times \Delta^d \to \mathbb{R}$ as **calibration estimation function**. We now propose a risk, which quantifies how close such a $h$ is to $h^*$. To achieve this, we slightly modify the mean-squared error loss function of Equation (1) in the following.

**Definition 1.** *For a prediction $c \in \mathbb{R}$, a target product measure $\mathbb{P}_Y \otimes \mathbb{P}_V$, and constants $p, p' \in \Delta^d$ we define the **calibration estimator loss** by*

$$L_{\mathrm{CE}}(c, \mathbb{P}_Y \otimes \mathbb{P}_V; p, p') := \mathbb{E}_{(Y,V) \sim \mathbb{P}_Y \otimes \mathbb{P}_V}\left[\left(\langle p - e_Y, p' - e_V \rangle - c\right)^2\right], \tag{14}$$

*where $e_i$ refers to the unit vector with a 1 at index $i$.*

Similar to the mean-squared error, it holds that $L_{\text{CE}}$ has an unique minimizer given by

$$\langle p - \mathbb{P}_Y, p' - \mathbb{P}_V \rangle = \arg\min_{c \in \mathbb{R}} L_{\text{CE}}\left(c, \mathbb{P}_Y \otimes \mathbb{P}_V; p, p'\right). \tag{15}$$

We use this definition of a novel loss to define the respective risk in the following.

**Definition 2.** *For a calibration estimator function $h\colon \Delta^d \times \Delta^d \to \mathbb{R}$, we define the **calibration estimation risk** by*

$$\mathscr{R}_{\text{CE}}(h) \coloneqq \mathbb{E}_{X,X'}\left[L_{\text{CE}}\left(h\left(f(X), f(X')\right), \mathbb{P}_{Y|f(X)=f(X)} \otimes \mathbb{P}_{Y|f(X)=f(X')}; f(X), f(X')\right)\right] \tag{16}$$

*with $X, X' \overset{iid}{\sim} \mathbb{P}_X$.*

Similar to $\mathscr{R}_{\text{MSE}}$ in Equation (3), we may also express $\mathscr{R}_{\text{CE}}$ in a simpler form, since it holds

$$\mathscr{R}_{\text{CE}}(h) = \mathbb{E}_{X,X',Y,Y'}\left[\left(\langle f(X) - e_Y, f(X') - e_{Y'}\rangle - h\left(f(X), f(X')\right)\right)^2\right] \tag{17}$$

with $(X, Y), (X', Y') \overset{iid}{\sim} \mathbb{P}_{XY}$. This formulation will be used in a later section to construct the empirical risk. Next, we establish that our proposed risk can distinguish the right solution almost surely.

**Theorem 1.** *For any $h\colon \Delta^d \times \Delta^d \to \mathbb{R}$ for which $h = h^*$ does **not** hold $\mathbb{P}_{f(X)} \otimes \mathbb{P}_{f(X)}$-almost surely we have that*

$$\mathscr{R}_{\text{CE}}(h) > \mathscr{R}_{\text{CE}}\left(h^*\right). \tag{18}$$

This property would be sufficient in practice, if we were using $h$ with arguments sampled from $\mathbb{P}_{f(X)} \otimes \mathbb{P}_{f(X)}$ to estimate the calibration error. However, as demonstrated by Equation (13), we predict $\text{CCE}_2$ via the same sample in both arguments. Mirroring the arguments results in a possible $\mathbb{P}_{f(X)} \otimes \mathbb{P}_{f(X)}$-null set since only elements in the diagonal $\mathscr{D}\left(\Delta^d\right) \coloneqq \left\{(p, p) \mid p \in \Delta^d\right\} \subset \Delta^d \times \Delta^d$ are considered. Consequently, the diagonal of an optimum $h'^*$ identified via $\mathscr{R}_{\text{CE}}$ may "slip through" the $\mathbb{P}_{f(X)} \otimes \mathbb{P}_{f(X)}$-a.s. guarantee in Theorem 1, and in turn may result in $\sqrt{\mathbb{E}\left[h'^*\left(f(X), f(X)\right)\right]} \neq \text{CCE}_2(f)$. To avoid such theoretical exceptions, we require additional theoretical assumptions, which we state in the following.

**Theorem 2.** *Assume a function $h\colon \Delta^d \times \Delta^d \to \mathbb{R}$ is continuous in all points of the diagonal $\mathscr{D}\left(\Delta^d \setminus A\right)$ with $A$ being a $\mathbb{P}_{f(X)}$-null set, and the target as a function $\mathbb{P}_{Y|f(X)}\colon \Delta^d \to \Delta^d$ is continuous $\mathbb{P}_{f(X)}$-almost surely. Further, assume the boundary of the support of $\mathbb{P}_{f(X)}$ does not involve a singular distribution, then it holds that*

$$\mathscr{R}_{\text{CE}}(h) = \mathscr{R}_{\text{CE}}\left(h^*\right) \implies \sqrt{\mathbb{E}_X\left[h\left(f(X), f(X)\right)\right]} = \text{CCE}_2(f). \tag{19}$$

This result states under which conditions we can expect that an optimal risk indicates a truthful calibration estimation. We briefly discuss these conditions, which are of purely technical nature and should not influence practical results. First, the continuity of $h$ is non-problematic since it is user-defined and infinitely many points of discontinuity are allowed (as long as they have no probability mass). The continuity of $\mathbb{P}_{Y|f(X)}$ may be considered the most relevant condition in practice, since we usually do not know about the nature of the target distribution. However, again, infinitely many points of discontinuity are allowed, as long as their overall probability mass is zero. The last assumption, namely that the boundary of the support is not allowed to be part of a singular distribution, disqualifies certain theoretically crafted distributions. One such example is the Cantor distribution, which consists of infinitely many disconnected points each of zero probability (Teschl, 2014). In summary, the purpose of the conditions in Theorem 2 is to guarantee that samples from $\mathbb{P}_{f(X)} \otimes \mathbb{P}_{f(X)}$ can be arbitrarily close to the diagonal $\mathscr{D}\left(\Delta^d\right)$ and that this closeness indicates how well $h$ matches $h^*$ on $\mathscr{D}\left(\Delta^d\right)$.

**Remark.** *Alternatively to our approach, one might also use a loss for finding a probabilistic model $\hat{g}(p) \approx \mathbb{P}_{Y|f(X)=p}$ and then use $\hat{h}(p, p') \coloneqq \langle p - \hat{g}(p), p' - \hat{g}(p')\rangle \approx \langle p - \mathbb{P}_{Y|f(X)=p}, p' - \mathbb{P}_{Y|f(X)=p'}\rangle$ as a solution. However, learning a predictive space $\Delta^d$ becomes increasingly more difficult for higher dimensions $d$ than a regression problem in $\mathbb{R}$. For example, our approach is invariant to any orthogonal matrix $M$ since $\langle Mx, My\rangle = \langle x, y\rangle$.*

### 3.2 Estimating Calibration for Finite Data

In this section, we propose a novel calibration evaluation pipeline for the finite data regime according to our theory. First, we give an unbiased and consistent estimator of the risk in form of an U-statistic. Then, we mimic the training-validation-testing pipeline of a conventional machine learning model for a debiased calibration estimate. This procedure allows to select between different calibration estimators and optimize their hyperparameters.

#### 3.2.1 Risk Estimator

Note that the risk as formulated in Equation (17), is an expectation of two i.i.d. tuples of random variables $(X, Y)$ and $(X', Y')$. Consequently, given an i.i.d. dataset $(X_1, Y_1), \ldots, (X_n, Y_n) \sim \mathbb{P}_{XY}$, we can construct an U-statistic estimator (Shao, 2003) via

$$\hat{\mathscr{R}}_{\mathrm{CE}}(h) := \frac{1}{n(n-1)} \sum_{i=1}^{n} \sum_{\substack{j=1 \\ i \neq j}}^{n} \left( \left\langle f(X_i) - e_{Y_i}, f(X_j) - e_{Y_j} \right\rangle - h\left(f(X_i), f(X_j)\right) \right)^2. \tag{20}$$

It holds that $\mathbb{E}\left[\hat{\mathscr{R}}_{\mathrm{CE}}(h)\right] = \mathscr{R}_{\mathrm{CE}}(h)$, and $\hat{\mathscr{R}}_{\mathrm{CE}}(h) \to \mathscr{R}_{\mathrm{CE}}(h)$ in distribution if $n \to \infty$. The estimator has quadratic complexity in $n$. However, an estimator with linear complexity can be constructed as well by excluding certain index combinations.

#### 3.2.2 Calibration-Evaluation Pipeline

In general, we cannot expect to find $h^*$. However, the empirical risk allows us to find a $\hat{h}_\eta \in H_\eta$ close to $h^*$, where $H_\eta$ is a model class with hyperparameter $\eta \in \Theta$ and search space $\Theta$. Examples for $\eta$ are the number of bins in Equation (10) or the bandwidth in Equation (11). Consequently, similar to traditional machine learning, we need a training-validation-test split to achieve a debiased estimation of the calibration error once we optimized $\eta$. Specifically, we use a training set to fit $\hat{h}_\eta$ and a validation set to find an optimal $\eta_{\mathrm{val}}$. The final calibration estimate is then computed by

$$\widehat{\mathrm{CE}}_2(f) := \sqrt{\frac{1}{n_{\mathrm{te}}} \sum_{X \in D_{\mathrm{te}}} \hat{h}_{\eta_{\mathrm{val}}}(f(X), f(X))}, \tag{21}$$

where $D_{\mathrm{te}}$ is the test set of size $n_{\mathrm{te}}$, and $\widehat{\mathrm{CE}}_2$ is representative for different notions of calibration. We may also use multiple training and validation sets via cross-validation, similar to conventional hyperparameter optimization (Bischl et al., 2023). Our proposed pipeline for estimating the calibration error of a classifier via cross-validation is presented in Algorithm 1. The algorithm mimics hyperparameter optimization of a conventional machine learning model (Bischl et al., 2023). Consequently, using our pipeline also implies a computational complexity of $O(k|\Theta|)$, where $k$ is the number of folds in cross-validation. In comparison, using a default hyperparameter, as in (Guo et al., 2017) or (Minderer et al., 2021), has $O(1)$ complexity since no optimization happens.

**Remark.** *We recommend not comparing the holdout test set risk of different calibration estimation functions since this would put a bias on the final calibration estimation. Neglecting this is similar to selecting an optimal classifier based on the test accuracy in a classification task.*

In this section, we have established our optimization and evaluation pipeline based on calibration estimation functions. Next, we show how already existing calibration error estimators used in the literature can be framed as calibration estimation functions.

## 4 Calibration Error Estimators as Calibration Estimation Functions

In this section, we first formulate the calibration estimation functions implicitly used in the literature. Then, we introduce two novel calibration estimators based on kernel ridge regression, which minimize regularized

---

**Algorithm 1** Evaluating the calibration of a given classifier and dataset by optimizing the calibration estimator. The evaluation dataset is split into a holdout set for estimating the calibration error, and another set, which is used for optimizing the calibration estimator via cross-validation.

---

**Input:** Evaluation dataset $D = \{(X_1, Y_1), \ldots, (X_n, Y_n)\}$, classifier $f$, model class $H_\eta$ of calibration estimation functions with hyperparameter search space $\Theta \ni \eta$, $k$ number of folds.
$D_{\mathrm{pr}} \leftarrow \{(f(X_1), Y_1), \ldots, (f(X_n), Y_n)\}$  ▷ compute classifier predictions
$D_{\mathrm{opt}}, D_{\mathrm{te}} \leftarrow \mathrm{split}(D_{\mathrm{pr}})$  ▷ create optimization and holdout test set
$\{(D_1^{\mathrm{train}}, D_1^{\mathrm{eval}}), \ldots, (D_k^{\mathrm{train}}, D_k^{\mathrm{eval}})\} \leftarrow \mathrm{CVfolds}(D_{\mathrm{opt}})$  ▷ create cross-validation folds
**for** $i = 1, \ldots, k$ **do**
   **for** $\eta \in \Theta$ **do**
      $\hat{h}_\eta^i \leftarrow \mathrm{fit}(H_\eta, D_i^{\mathrm{train}})$  ▷ fit for a given hyperparameter and training set
      $\mathrm{risk}_\eta^i \leftarrow \hat{\mathscr{R}}_{\mathrm{CE}}(\hat{h}_\eta^i)$  ▷ compute risk for $\hat{h}_\eta$ with data $D_i^{\mathrm{eval}}$ according to Eq. (20)
   **end for**
**end for**
$\eta_{\mathrm{val}} \leftarrow \arg\min_{\eta \in H} \frac{1}{k} \sum_{i=1}^k \mathrm{risk}_\eta^i$  ▷ get the hyperparameter with the smallest average risk
Define $\hat{h}_{\eta_{\mathrm{val}}}(x, y) := \frac{1}{k} \sum_{i=1}^k \hat{h}_{\eta_{\mathrm{val}}}^i(x, y)$
Compute $\widehat{\mathrm{CE}}_2(f)$ with $\hat{h}_{\eta_{\mathrm{val}}}$ and $D_{\mathrm{te}}$ according to Eq. (21)
**return** $\widehat{\mathrm{CE}}_2(f)$

---

versions of the empirical risk in Equation (20). All calibration estimation functions can be seen as preliminary to future research, since we may find function classes with lower validation risk by expanding the search space and computational resources. All missing proofs are located in Appendix C.

## 4.1 Reframing Binning and Kernel Density based Calibration Estimators

In this section, we show that the binning estimator $\mathrm{TCE}_2^{\mathrm{bin}}$ of Equation (10) and the kernel density ratio estimator $\mathrm{CCE}_2^{\mathrm{kde}}$ of Equation (11) are the mean prediction of an implicit calibration estimator function of the form

$$\frac{1}{n} \sum_{i=1}^n h(f(X_i), f(X_i)) \tag{22}$$

for some $h \colon \Delta^d \times \Delta^d \to \mathbb{R}$ and an i.i.d. dataset $(X_1, Y_1), \ldots, (X_n, Y_n) \sim \mathbb{P}_{XY}$. For the binning-based estimator, define

$$h_{\mathrm{bin}}(p, p') := \left( \sum_{m=1}^M (\mathrm{conf}(B_m) - \mathrm{acc}(B_m)) \mathbf{1}_{p \in I_m} \right) \left( \sum_{m=1}^M (\mathrm{conf}(B_m) - \mathrm{acc}(B_m)) \mathbf{1}_{p' \in I_m} \right), \tag{23}$$

where $I_1 \ldots I_M$, $\mathrm{acc}(B_m)$, and $\mathrm{conf}(B_m)$ are defined as above in Equation (10). It holds that

$$\left( \mathrm{TCE}_2^{\mathrm{bin}}(f) \right)^2 = \frac{1}{n} \sum_{i=1}^n h_{\mathrm{bin}}(f(X_i), f(X_i)). \tag{24}$$

Similarly, one may formulate the debiased (but not unbiased) estimator of Kumar et al. (2019).

Further, we can also put the estimator of Popordanoska et al. (2022b) in the form of Equation (22) by defining

$$h_{\mathrm{kde}}(p, p') := \left\langle p - \frac{\sum_{i=1}^n e_{Y_i} k_{\mathrm{dir}}(f(X_i), p)}{\sum_{i=1}^n k_{\mathrm{dir}}(f(X_i), p)}, p' - \frac{\sum_{i=1}^n e_{Y_i} k_{\mathrm{dir}}(f(X_i), p')}{\sum_{i=1}^n k_{\mathrm{dir}}(f(X_i), p')} \right\rangle, \tag{25}$$

where $k_{\mathrm{dir}}$ is the Dirichlet kernel as defined above in Equation (11). Again, it holds that

$$\left( \mathrm{CCE}_2^{\mathrm{kde}}(f) \right)^2 = \frac{1}{n} \sum_{i=1}^n h_{\mathrm{kde}}(f(X_i), f(X_i)). \tag{26}$$

We will also use an analogous estimator $\mathrm{TCE}_2^{\mathrm{kde}}$ for estimating $\mathrm{TCE}_2$ in the experiment sections. Extending the binning-based estimator to $\mathrm{CCE}_2$ is in general not possible. Further, our calibration evaluation pipeline allows us to compare the risk of different calibration estimators. This motivates the introduction of novel calibration estimators, especially for canonical calibration errors, to have a larger search space to optimize over. In the following, we introduce a novel class of calibration estimators based on kernel ridge regression, which minimizes the empirical risk in Equation (20) plus a regularization term under typical kernel ridge regression assumptions.

## 4.2 Novel Calibration Estimators based on Kernel Ridge Regression

Here, we propose two novel calibration estimators, which are derived as closed-form solutions under the typical kernel ridge regression assumptions, see, e.g., (Schölkopf & Smola, 2002; Bach, 2024). The following approach is based on the notion of ordinary Kronecker kernel ridge regression (Stock et al., 2018). Specifically, we require a reproducing kernel Hilbert space (RKHS) $\mathscr{H}$ with an associated feature map $\phi \colon \Delta^d \to \mathscr{H}$, kernel $k_{\mathscr{H}}$, inner product $\langle ., .\rangle_{\mathscr{H}}$, and norm $\|.\|_{\mathscr{H}}$. Denote with $\mathscr{H} \otimes \mathscr{H}$ the tensor product of the Hilbert space with itself, with respective feature map $(\phi \otimes \phi) \colon \Delta^d \times \Delta^d \to \mathscr{H} \otimes \mathscr{H}$. Next, assume that $\langle f(X) - e_Y, f(X') - e_{Y'}\rangle = \langle g^*, (\phi \otimes \phi)(p, p')\rangle_{\mathscr{H} \otimes \mathscr{H}} + \epsilon$ for some $g^* \in \mathscr{H} \otimes \mathscr{H}$ and zero-mean noise term $\epsilon$. Define the kernel ridge objective for a $g \in \mathscr{H} \otimes \mathscr{H}$ via

$$\hat{\mathscr{R}}_{\mathrm{CE},\lambda}(g) := \frac{1}{n^2} \sum_{i=1}^{n} \sum_{j=1}^{n} \left(\left\langle f(X_i) - e_{Y_i}, f(X_j) - e_{Y_j}\right\rangle - h\left(f(X_j), f(X_j)\right)\right)^2 + \lambda \|g\|_{\mathscr{H} \otimes \mathscr{H}}^2, \tag{27}$$

where $h(p, p') = \langle g, (\phi \otimes \phi)(p, p')\rangle_{\mathscr{H} \otimes \mathscr{H}}$. This objective is in essence the risk in Equation (20) plus a regularization constant. Then, a closed-form minimizer can be found, which results in the predictor

$$h_{\mathrm{kkr}}(p, p') := \mathrm{vec}^{\mathsf{T}} \left(\boldsymbol{\Delta}_{Yf(X)}^{\mathsf{T}} \boldsymbol{\Delta}_{Yf(X)}\right) \left(\mathbf{K}_{f(X)} \otimes \mathbf{K}_{f(X)} + \lambda n^2 I\right)^{-1} \left(\mathbf{k}_{f(X)}(p) \otimes \mathbf{k}_{f(X)}(p')\right), \tag{28}$$

where $\otimes$ becomes the Kronecker product, $\boldsymbol{\Delta}_{Yf(X)} := \left(f(X_1) - e_{Y_1} \;\; \cdots \;\; f(X_n) - e_{Y_n}\right) \in \mathbb{R}^{d \times n}$, $\mathbf{K}_{f(X)} := \left(k_{\mathscr{H}}\left(f(X_i), f(X_j)\right)\right)_{i,j \in \{1\ldots n\}} \in \mathbb{R}^{n \times n}$, and $\mathbf{k}_{f(X)}(p) := \left(k_{\mathscr{H}}(f(X_i), p)\right)_{i \in \{1\ldots n\}} \in \mathbb{R}^n$. Without further modifications, computing Equation (28) has runtime complexity $O(n^6)$ due to the matrix inverse, which is practically infeasible. To reduce the complexity, we classically for Kronecker products make use of the eigenvalue decomposition $\mathbf{K}_{f(X)} = Q_{f(X)} \mathrm{diag}(\lambda_1, \ldots, \lambda_n) Q_{f(X)}^{\mathsf{T}}$, which is in $O(n^3)$. Define $\tilde{\Lambda}_{f(X)} \in \mathbb{R}^{n \times n}$ with $\left[\tilde{\Lambda}_{f(X)}\right]_{ij} = \frac{1}{\lambda_i \lambda_j + \lambda n^2}$ and denote with $\odot$ the Hadamard product. It holds that

$$h_{\mathrm{kkr}}(p, p') = \mathbf{k}_{f(X)}^{\mathsf{T}}(p) Q_{f(X)} \left(\tilde{\Lambda}_{f(X)} \odot Q_{f(X)}^{\mathsf{T}} \boldsymbol{\Delta}_{Yf(X)}^{\mathsf{T}} \boldsymbol{\Delta}_{Yf(X)} Q_{f(X)}\right) Q_{f(X)}^{\mathsf{T}} \mathbf{k}_{f(X)}(p'). \tag{29}$$

This representation consists of multiplications of $n \times n$ matrices and in consequence is in $O(n^3)$. A more general approach uses the Schur decomposition to achieve such a reduction in complexity (Moravitz Martin & Van Loan, 2007). Naively computing the empirical generalization error $\hat{\mathscr{R}}_{\mathrm{CE}}(h_{\mathrm{kkr}})$ on an evaluation set $(X_1', Y_1'), \ldots, (X_{n'}', Y_{n'}')$ for some $n' \propto n$ has complexity $O(n^5)$, which is again prohibitive in practice. However, we can also reduce this complexity to $O(n^3)$ since it holds

$$\hat{\mathscr{R}}_{\mathrm{CE}}(h_{\mathrm{kkr}}) = \frac{1}{n'(n'-1)} \sum_{i=1}^{n'} \sum_{\substack{j=1 \\ j \neq i}}^{n'} \left(\left\langle f(X_i) - e_{Y_i}, f(X_j) - e_{Y_j}\right\rangle - H_{ij}^{\mathrm{kkr}}\right)^2, \tag{30}$$

with

$$H^{\mathrm{kkr}} := \mathbf{K}_{f(X)f(X')}^{\mathsf{T}} Q_{f(X)} \left(\tilde{\Lambda}_{f(X)} \odot Q_{f(X)}^{\mathsf{T}} \boldsymbol{\Delta}_{Yf(X)}^{\mathsf{T}} \boldsymbol{\Delta}_{Yf(X)} Q_{f(X)}\right) Q_{f(X)}^{\mathsf{T}} \mathbf{K}_{f(X)f(X')} \in \mathbb{R}^{n' \times n'} \tag{31}$$

and $\left[\mathbf{K}_{f(X)f(X')}\right]_{ij} := k_{\mathscr{H}}\left(f(X_i), f(X_j')\right)$.

Alternatively, one may also fit a kernel ridge regressor for the problem assumption $f(X) - e_Y = \tilde{g}^* \phi(X) + \epsilon$ with $\tilde{g}^* \in \left\{\tilde{g} \colon \mathscr{H} \to \Delta^d\right\}$, and then use the result as plug-in for the inner product. This is referred to as

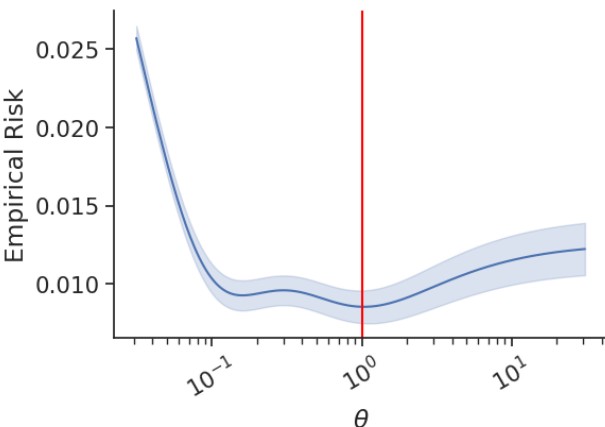

Figure 1: Simulated experiment for estimating $CCE_2$ in a task with 5 classes and 500 instances. The empirical risk correctly identifies the ideal calibration estimator with $\theta = 1$ indicated by the red line. Standard deviations across multiple seeds indicate the empirical risk stability.

two-step kernel ridge regression (Stock et al., 2018) or U-statistic regression (Park et al., 2021). The model then becomes

$$h_{\text{ukkr}}\left(p, p'\right) := \mathbf{k}_{f(X)}^{\top}\left(p\right)\left(\mathbf{K}_{f(X)} + \lambda nI\right)^{-1}\boldsymbol{\Delta}_{Yf(X)}^{\top}\boldsymbol{\Delta}_{Yf(X)}\left(\mathbf{K}_{f(X)} + \lambda nI\right)^{-\top}\mathbf{k}_{f(X)}\left(p'\right). \tag{32}$$

It holds that $h_{\text{ukkr}} = h_{\text{kkr}}$ if the kernel used for $h_{\text{kkr}}$ incorporates the regularisation constant $\lambda$ or when $\lambda = 0$ (Stock et al., 2018). We define the calibration estimators based on kernel ridge regression for $CCE_2$ as

$$\text{CCE}_2^{\text{kkr}}\left(f\right) := \sqrt{\frac{1}{n}\sum_{i=1}^{n} h_{\text{kkr}}\left(f\left(X_i\right), f\left(X_i\right)\right)} \tag{33}$$

and

$$\text{CCE}_2^{\text{ukkr}}\left(f\right) := \sqrt{\frac{1}{n}\sum_{i=1}^{n} h_{\text{ukkr}}\left(f\left(X_i\right), f\left(X_i\right)\right)}. \tag{34}$$

We also use analogous estimators $\text{TCE}_2^{\text{kkr}}$ and $\text{TCE}_2^{\text{ukkr}}$ for estimating $\text{TCE}_2$.

We have now established a large pool of possible calibration estimation functions we can compare and optimize. In the next section, we perform top-label confidence and canonical calibration evaluations with the discussed and proposed estimators. Specifically, we use the proposed calibration estimation risk in Equation (20) for comparison and the proposed calibration-evaluation pipeline of Section 3.2.2 for calibration estimation.

## 5 Experiments

In this section, we demonstrate how to use our proposed risk framework in practice. We first run a simulation with known ground truth. Then, we evaluate the risk of the different calibration estimation functions defined in Section 4 and the respective estimated calibration error across a variety of image classification datasets and models. We focus on top-label confidence since it is the most prominent, and canonical calibration, which is the most general. The source code is publicly available at `https://github.com/SebGGruber/Optimizing_Calibration_Estimators`.

### 5.1 Simulation

We construct a simulation experiment with known ground truth to demonstrate how our proposed risk identifies the solution. For this, we draw i.i.d. samples $P_1, \ldots, P_{500} \sim \text{Dir}\left(\alpha_1, \ldots, \alpha_5\right)$ from a Dirichlet distribution with

Table 1: Validation set square root risk $\sqrt{\hat{\mathscr{R}}_{\text{CE}}} \times 100$ of $\text{TCE}_2$ estimators for CIFAR10 models with optimized hyperparameters. Lower is better. The estimator $\text{TCE}_2^{\text{kde}}$ performs worse and $\text{TCE}_2^{\text{ukkr}}$ better or equal than the other estimators. The large risk of $\text{TCE}_2^{\text{kde}}$ translates to an outlier calibration estimation in Figure 2a.

| Model Estimator | LeNet-5 | Densenet-40 | ResNetWide-32 | Resnet-110 | Resnet-110 SD |
|---|---|---|---|---|---|
| $\text{TCE}_2^{15-bins}$ | 14.96 ± 0.31 | 6.14 ± 0.12 | 5.03 ± 0.2 | 5.4 ± 0.24 | 4.73 ± 0.25 |
| $\text{TCE}_2^{bins}$ | 14.96 ± 0.31 | 6.12 ± 0.12 | 5.03 ± 0.2 | 5.39 ± 0.23 | 4.72 ± 0.25 |
| $\text{TCE}_2^{kde}$ | 14.98 ± 0.31 | 13.7 ± 0.27 | 12.31 ± 0.65 | 7.46 ± 0.35 | 10.65 ± 0.58 |
| $\text{TCE}_2^{kkr}$ | 14.96 ± 0.31 | 6.13 ± 0.12 | 5.03 ± 0.2 | 5.39 ± 0.24 | 4.72 ± 0.25 |
| $\text{TCE}_2^{ukkr}$ | 14.96 ± 0.31 | 6.11 ± 0.12 | 5.03 ± 0.2 | 5.38 ± 0.24 | 4.72 ± 0.25 |

Table 2: Validation set risk $\sqrt{\hat{\mathscr{R}}_{\text{CE}}} \times 100$ of $\text{TCE}_2$ for Cifar100 models. Lower is better. Similar as before, the estimator $\text{TCE}_2^{\text{kde}}$ performs worse than the other estimators except for LeNet-5. The best performing are $\text{TCE}_2^{\text{ukkr}}$ and $\text{TCE}_2^{\text{bins}}$.

| Model Estimator | LeNet-5 | Densenet-40 | ResNetWide-32 | Resnet-110 | Resnet-110 SD |
|---|---|---|---|---|---|
| $\text{TCE}_2^{15-bins}$ | 18.51 ± 0.16 | 20.66 ± 0.40 | 18.40 ± 0.19 | 18.67 ± 0.43 | 17.18 ± 0.20 |
| $\text{TCE}_2^{bins}$ | 18.50 ± 0.16 | 20.43 ± 0.38 | 18.24 ± 0.17 | 18.61 ± 0.42 | 17.11 ± 0.20 |
| $\text{TCE}_2^{kde}$ | 18.50 ± 0.16 | 23.74 ± 0.41 | 23.28 ± 0.18 | 19.69 ± 0.47 | 18.21 ± 0.19 |
| $\text{TCE}_2^{kkr}$ | 18.50 ± 0.16 | 20.52 ± 0.39 | 18.29 ± 0.18 | 18.59 ± 0.42 | 17.11 ± 0.20 |
| $\text{TCE}_2^{ukkr}$ | 18.50 ± 0.16 | 20.51 ± 0.40 | 18.28 ± 0.18 | 18.61 ± 0.43 | 17.12 ± 0.21 |

concentration parameters $\alpha_1 = \cdots = \alpha_5 = 0.04$. These samples represent the (in practice unknown) ground truth probability vectors of a classification task with 500 instances and 5 classes. Then, we sample $Y_i \sim P_i$ for $i = 1, \ldots 500$ as target labels. We set the hypothetical model predictions to $f(X_i) = \text{softmax}\left(\frac{3}{10}\log P_i\right)$ for $i = 1, \ldots 500$, which represent miscalibrated predictions. The concentration parameters were set such that the model would have an accuracy of $\approx 90\%$. We then define a calibration estimation function $h_{\text{sim}}(p, p') := \left\langle p - \text{softmax}\left(\frac{10}{3}\theta \log p\right), p' - \text{softmax}\left(\frac{10}{3}\theta \log p'\right)\right\rangle$, which has a single learnable parameter $\theta \in \mathbb{R}$ referred to as temperature. The estimation function was chosen such that it matches the ground truth $h_{\text{sim}} = h^*$ if and only if $\theta = 1$. In Figure 1, we plot the mean results with standard deviations of the empirical risk according to 100 repetitions of the experiment. As can be seen, the empirical risk correctly identifies the correct calibration estimation function with $\theta = 1$.

## 5.2 Real World Settings

In the following, we evaluate and compare estimators discussed in Section 4 according to our novel calibration-evaluation pipeline introduced in Section 3.2.2 on standard image classifier setups, like CIFAR and ImageNet.

## Technical Setup

The experiments are conducted across several model-dataset combinations, whose logit sets are openly accessible (Kull et al., 2019; Rahimi et al., 2020; Gruber & Buettner, 2022).[1] The image classification datasets in use are CIFAR10 with 10 classes, CIFAR100 with 100 classes (Krizhevsky, 2009), and ImageNet with 1,000 classes (Deng et al., 2009). Since we restrict ourselves to evaluating the calibration error estimate of

---

[1] https://github.com/MLO-lab/better_uncertainty_calibration

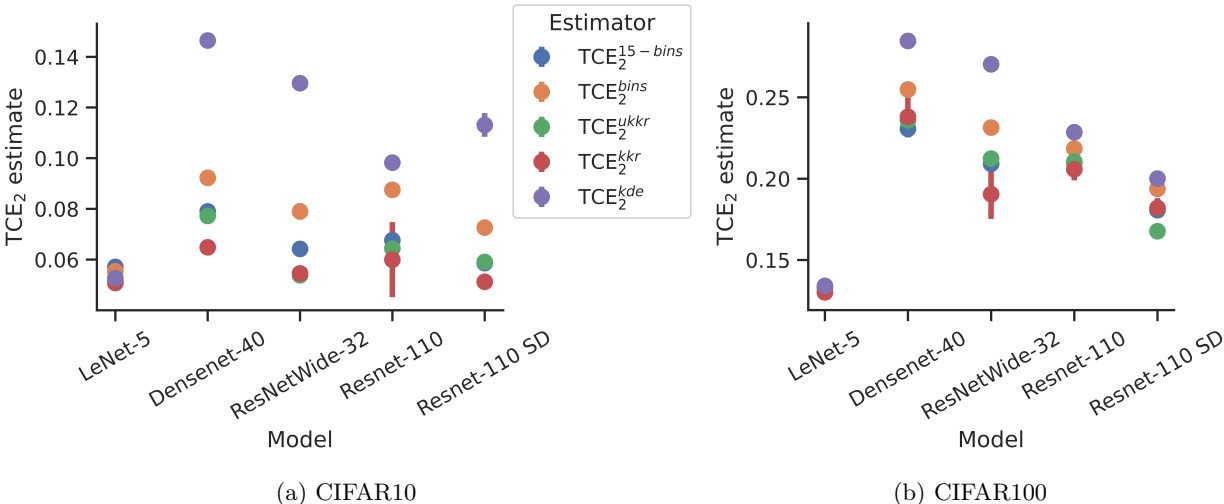

Figure 2: Different $\mathrm{TCE}_2$ estimates of different models. Most calibration estimates approximately agree with each other. Only $\mathrm{TCE}_2^{\mathrm{kde}}$ is an outlier for Densenet-40, ResNetWide-32, and Resnet-110 SD. However, it also shows an increased calibration estimation risk in these cases (c.f. Table 1).

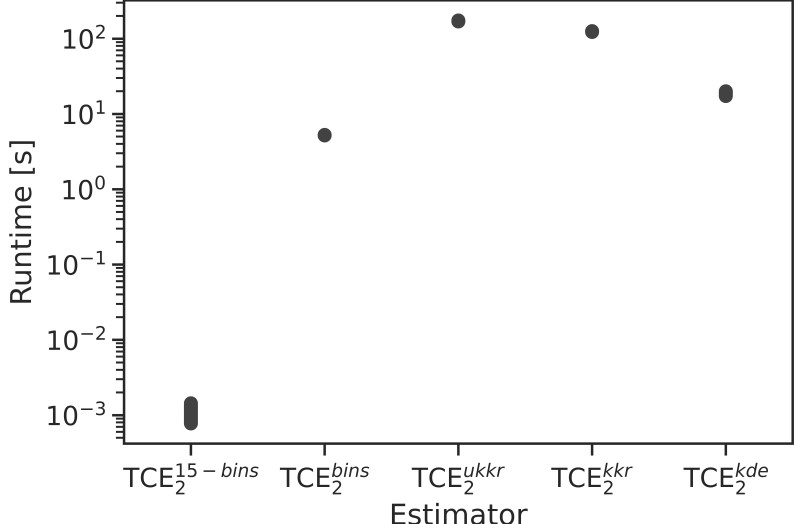

Figure 3: Average runtime with error bars of the estimators in Figure 2a on a single CPU thread. Optimizing the hyperparameters has a substantial computational cost.

the models, we only use the test set of each dataset (CIFAR: $n = 10{,}000$, ImageNet: $n = 25{,}000$). Modifying or selecting models based on the calibration estimate would require using the validation set instead. The included models are LeNet-5 (LeCun et al., 1998), ResNet-110, ResNet-110 SD, ResNet-152 (He et al., 2016), Wide ResNet-32 (Zagoruyko & Komodakis, 2016), DenseNet-40, DenseNet-161 (Huang et al., 2017), and PNASNet-5 Large (Liu et al., 2018). We did not conduct model training ourselves and refer to Kull et al. (2019) and Rahimi et al. (2020) for further details. We evaluate top-label confidence calibration and canonical calibration estimators for these classifiers.

We run the calibration-evaluation pipeline proposed in Section 3.2.2 with a random split of the original test set, using 80% for tuning the calibration estimator function via cross-validation and 20% for the calibration test set $\mathscr{D}_{\mathrm{te}}$, which computes the mean in Equation (21). In all experiments, we use 5-fold cross-validation to

optimize the hyperparameters of a calibration estimator function. For the best performing hyperparameter, the models across all folds are used as an ensemble predictor for the final calibration estimation on the set $\mathscr{D}_{\mathrm{te}}$. This approach also allows us to include error bars according to the cross validation folds.

As calibration estimator functions, we consider $h_{\mathrm{bin}}$, $h_{\mathrm{kde}}$, $h_{\mathrm{kkr}}$, and $h_{\mathrm{ukkr}}$ for top-label confidence, as well as $h_{\mathrm{kde}}$, $h_{\mathrm{kkr}}$, and $h_{\mathrm{ukkr}}$ canonical calibration. For both kernel ridge regression models, we use the RKHS of the RBF kernel $k_{\mathrm{rbf}}(x,y) = \exp\left(-\gamma \|x-y\|^2\right)$. We set $\gamma = \frac{1}{2}$ based on preliminary evaluations. We also evaluate $h_{\mathrm{bin}}$ with 15 bins without hyperparameter optimization, which we refer to as $\mathrm{TCE}_2^{15-\mathrm{bins}}$. This corresponds to a common default choice in current practice (Guo et al., 2017; Detlefsen et al., 2022). More details on the hyperparameter search spaces are given in Appendix B.

**Results**

We now discuss the experimental results. All reported risks are with respect to the holdout sets in the cross-validation folds. The reported calibration estimations are with respect to the calibration test set (20% of the original test set). The error bars for the risk and calibration estimations are the standard errors according to the cross-validation folds. In Table 1 we show the performance across different models of CIFAR10, and in Table 2 across models of CIFAR100. Here, we compare the calibration estimation functions for top-label confidence calibration. As can be seen, no calibration estimation function dominates all others. Specifically, $\mathrm{TCE}_2^{\mathrm{kde}}$ performs worst for all models, even when we consider the error bars. The estimator $\mathrm{TCE}_2^{\mathrm{ukkr}}$ outperforms the other estimators, however, the difference is too marginal with respect to the error bars to come to a confident conclusion. For the CIFAR100 models, $\mathrm{TCE}_2^{\mathrm{kde}}$ performs more similar to the other estimators. Further, the optimized binning estimator $\mathrm{TCE}_2^{\mathrm{bins}}$ shows the strongest performance and not $\mathrm{TCE}_2^{\mathrm{ukkr}}$ anymore. However, again, the error bars are too large to designate a definitive ranking. Further, for the LeNet-5 model in CIFAR10 and CIFAR100, our risk is not sufficiently sensitive to rank the estimators.

In Figure 2 we depict the corresponding $\mathrm{TCE}_2$ estimations. As can be seen, only $\mathrm{TCE}_2^{\mathrm{kde}}$ is occasionally an outlier relative to the other estimators, which is expected based on the reported risk values of Table 1 and Table 2. Even though, we cannot spot a direct connection between all risk values and the estimated calibration values in Figure 2, the large risk of $\mathrm{TCE}_2^{\mathrm{kde}}$ is indicative of a worse estimation. However, it is not surprising that differences in the risk do not always translate to differences in the estimated values, since the loss does not measure in which direction a calibration estimator function gives wrong predictions. Figure 3 shows the wall-clock runtime of each estimator in Figure 2a averaged across the classifiers for CIFAR10. Since all estimators only use the classifier outputs, the classifiers have no systematic influence on the runtime. As can be seen, using our optimization pipeline adds substantial computational costs to the calibration estimation. This is analogous to the costs of hyperparameter optimization for conventional machine learning models (Bischl et al., 2023).

In Table 3, we report the risk values of the canonical calibration estimator functions for CIFAR100 classifiers. As can be seen, $\mathrm{CCE}_2^{\mathrm{kde}}$ performs better than in previous results, outperforming the other approaches in some cases. We can also see that $\mathrm{CCE}_2^{\mathrm{ukkr}}$ performs better than $\mathrm{CCE}_2^{\mathrm{kkr}}$, which is a continuous trend across all results. However, the error bars dominate the performance difference and no clear cut conclusion can be made. In Figure 4, we show the respective calibration estimates.

In Appendix B, we offer additional results regarding top-label confidence calibration for ImageNet classifiers and canonical calibration for CIFAR10 classifiers. In summary, no calibration estimator outperforms the other approaches across all settings. Additionally, risk performance is often indicative of outlying calibration estimates. This underlines the requirement of a risk to assess which estimator to use for evaluating the calibration of a new model in practice. We may expect to find better estimators by extending the search space (e.g., by considering different kernels), or by including other model classes, like boosted trees or neural networks (Bishop & Nasrabadi, 2006). However, the proposed risk may not be sufficiently sensitive to rank the estimators according to their performance. Future research may involve exploring alternative loss functions for more sensitive results.

Table 3: Validation set square root risk $\sqrt{\hat{\mathscr{R}}_{\mathrm{CE}}} \times 100$ of $CCE_2$ estimators for CIFAR100 models. Again, lower is better. Contrary to previous results, the estimator $CCE_2^{\mathrm{kde}}$ manages to outperform the kernel ridge regression based estimators in some scenarios.

| Model
Estimator | LeNet-5 | Densenet-40 | ResNetWide-32 | Resnet-110 | Resnet-110 SD |
|---|---|---|---|---|---|
| $CCE_2^{kde}$ | 8.38 ± 0.06 | 5.61 ± 0.09 | 4.98 ± 0.05 | 5.14 ± 0.1 | 4.79 ± 0.03 |
| $CCE_2^{kkr}$ | 8.36 ± 0.06 | 5.56 ± 0.09 | 5.00 ± 0.05 | 5.16 ± 0.1 | 4.80 ± 0.03 |
| $CCE_2^{ukkr}$ | 8.35 ± 0.06 | 5.56 ± 0.09 | 5.01 ± 0.05 | 5.16 ± 0.1 | 4.80 ± 0.02 |

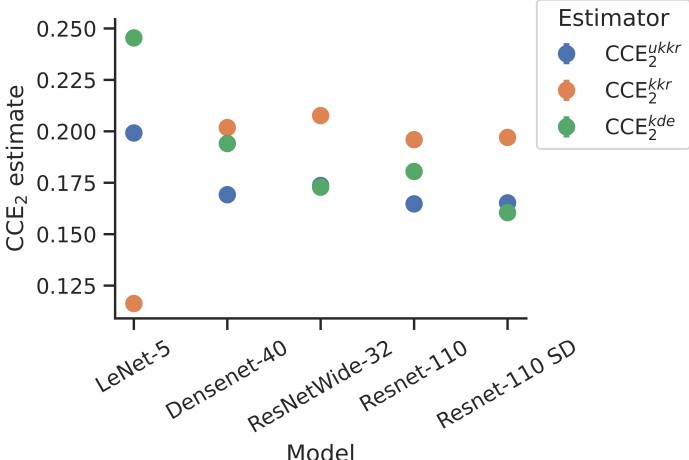

Figure 4: Different $CCE_2$ estimates for CIFAR100 models. The risk values of Table 3 do not relate to the calibration estimate but only indicate which estimator to trust more (here: $CCE_2^{\mathrm{kde}}$).

## 6 Conclusion

In this work, we introduced a mean-squared error based risk to compare different calibration estimators. This is the first approach in the literature to compare different calibration estimators on real-world datasets. We offer measure theoretic conditions for when the risk identifies an ideal estimator. We also derive novel calibration estimators as closed-form minimizers of the empirical risk based on kernel ridge regression assumptions. Further, using an empirical risk enables to perform hyperparameter optimization and estimator selection via a training-validation-testing pipeline, similar to conventional machine learning. In the experiments, we optimize the hyperparameters of common calibration estimators in the literature on popular real-world benchmarks, and compare the risks of different optimized estimators. No dominating calibration estimator was found, which emphasises the requirement of using our risk to detect an appropriate estimator for new settings in practice.

**Acknowledgments**

The authors are very thankful to Achim Hekler, who computed the logits for the VisionTransformer experiments in this work. Further, the authors would like to thank David Holzmüller, Eugène Berta, and Florian Buettner for fruitful discussions regarding this work.

The majority of this work was conducted during a research visit at Inria, which was partly funded by a "DKFZ Cancer Research Academy Short-term Research Visit Grant".

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

## A  Overview

Here, we first give additional experimental content in Appendix B. The missing proofs are located in Appendix C.

## B  Extended Experiments

In the following, we discuss further experimental details and results. Specifically, we evaluate the calibration of VisionTransformer classifiers (Dosovitskiy et al., 2020) trained on MedMNIST datasets (Yang et al., 2021) according to our approach.

**Additional Details**

We use the implementation of the calibration estimator function $h_{\mathrm{kde}}$ given by the original authors (Popordanoska et al., 2022b). For a small fraction of inputs, this implementation returns NaN as the prediction. We remove these instances from the risk and calibration estimation calculation of $h_{\mathrm{kde}}$, which has a neglectable effect.

As hyperparameter search spaces for the TCE experiments, we consider $\{5i \mid i = 1, \ldots, 20\}$ for the number of bins in $h_{\mathrm{bin}}$, a bandwidth in $\left\{10^{-5(i-1)/14-(1-(i-1)/14))} \mid i = 1, \ldots, 15\right\} \cup \{0.2i \mid i = 1, \ldots, 5\}$ for the Dirichlet kernel of $h_{\mathrm{kde}}$ according to Popordanoska et al. (2022a), a regularization constant $\lambda \in \left\{n^{0.5} 10^{-2i+1} \mid i = 1, \ldots, 9\right\}$ for $h_{\mathrm{kkr}}$, and $\lambda \in \left\{n^{0.5} 10^{-i} \mid i = 1, \ldots, 9\right\}$ for $h_{\mathrm{ukkr}}$. For the CCE experiments, we consider the same set of bandwidths for the Dirichlet kernel of $h_{\mathrm{kde}}$, a regularization constant $\lambda \in \left\{n^{0.5} 10^{-i+9} \mid i = 1, \ldots, 18\right\}$ for $h_{\mathrm{kkr}}$, and $\lambda \in \left\{n^{0.5} 10^{-0.5i+4.5} \mid i = 1, \ldots, 18\right\}$ for $h_{\mathrm{ukkr}}$.

All experiments are run on an Intel(R) Xeon(R) Gold 5218R with 2.1 GHz and a Macbook Pro M1.

**Additional Results**

In the following, we discuss the risks and calibration estimations of some cases left out of the main paper and VisionTransformer classifiers.

In Table 4 we show the risk of the top-label confidence calibration estimators for ImageNet with various models. All calibration estimation functions show similar risk except $\mathrm{TCE}_2^{\mathrm{kde}}$, which is worse for DenseNet-161 and Resnet-152. This is in agreement with Figure 2b, where the estimated calibration values are also mostly similar. The risks in Table 5 for canonical calibration estimators in the case of CIFAR10 show slightly different results: Here, the risk fails to distinguish the performance between the different estimators. Only $\mathrm{CCE}_2^{\mathrm{kde}}$ outperforms the other approaches for Resnet-110.

We train the VisionTransformer architecture (Dosovitskiy et al., 2020) on the MedMNIST classification datasets Blood, OCT, and Derma (Yang et al., 2021). Specifically, we use a pre-trained classifier from Huggingface[2] and fine-tune with a modification of (Capelle, 2022). The respective risk values are in Table 6. The kernel density estimator performs worse than the other approaches. No definitive winner can be declared due to the wide error bounds. In Figure 6, we show the associated calibration estimates. Similar to before, the risk values do not link to the calibration estimates.

In summary, the results mimic the ones in the main paper and it is not apparent which estimator to use in practice without considering our proposed risk. However, the risk may be insensitive regarding the various estimator performances.

---

[2] https://huggingface.co/timm/vit_base_patch16_224.orig_in21k_ft_in1k (Accessed on 2nd Jan 2025)

Table 4: Validation set square root risk $\sqrt{\hat{\mathscr{R}}_{\mathrm{CE}}} \times 100$ of $\mathrm{TCE}_2$ for different ImageNet models. Lower is better. The estimator $\mathrm{TCE}_2^{\mathrm{kde}}$ performs worse than the other estimators for DenseNet-161 and ResNet-152. For Pnasnet-5, all estimators perform similarly.

| Model
Estimator | DenseNet-161 | Resnet-152 | Pnasnet-5 |
|---|---|---|---|
| $\mathrm{TCE}_2^{15-bins}$ | 12.17 ± 0.16 | 12.58 ± 0.14 | 10.63 ± 0.16 |
| $\mathrm{TCE}_2^{bins}$ | 12.17 ± 0.16 | 12.57 ± 0.14 | 10.63 ± 0.16 |
| $\mathrm{TCE}_2^{kde}$ | 12.31 ± 0.17 | 12.77 ± 0.14 | 10.63 ± 0.16 |
| $\mathrm{TCE}_2^{kkr}$ | 12.17 ± 0.16 | 12.57 ± 0.14 | 10.63 ± 0.16 |
| $\mathrm{TCE}_2^{ukkr}$ | 12.17 ± 0.16 | 12.57 ± 0.14 | 10.63 ± 0.16 |

Table 5: Validation set risk $\sqrt{\hat{\mathscr{R}}_{\mathrm{CE}}} \times 100$ of $\mathrm{CCE}_2$ for CIFAR10 models. Lower is better. All estimators perform similarly.

| Model
Estimator | LeNet-5 | Densenet-40 | ResNetWide-32 | Resnet-110 | Resnet-110 SD |
|---|---|---|---|---|---|
| $\mathrm{CCE}_2^{kde}$ | 13.53 ± 0.27 | 5.13 ± 0.13 | 4.22 ± 0.16 | 4.46 ± 0.14 | 3.89 ± 0.2 |
| $\mathrm{CCE}_2^{kkr}$ | 13.53 ± 0.27 | 5.13 ± 0.13 | 4.22 ± 0.16 | 4.47 ± 0.14 | 3.89 ± 0.2 |
| $\mathrm{CCE}_2^{ukkr}$ | 13.53 ± 0.27 | 5.13 ± 0.13 | 4.22 ± 0.16 | 4.47 ± 0.14 | 3.89 ± 0.2 |

## C  Missing Proofs

Here, we present the missing proofs of the main part. Specifically, we prove Theorem 1 in Section C.1, Theorem 2 in Section C.2, and various statements of Section 3.2 in Section C.3.

### C.1  Proof for Theorem 1

We show that $\mathscr{R}_{\mathrm{CE}}(h) > \mathscr{R}_{\mathrm{CE}}(h^*)$.

For this, we require that $\langle p - \mathbb{P}_Y, p' - \mathbb{P}_V \rangle$ is the unique minimizer of $L_{\mathrm{CE}}(., \mathbb{P}_Y \otimes \mathbb{P}_V; p, p')$, which holds since

$$
\begin{aligned}
&\frac{\partial}{\partial c} L_{\mathrm{CE}}\left(c, \mathbb{P}_Y \otimes \mathbb{P}_V; p, p'\right) \\
&= \frac{\partial}{\partial c} \mathbb{E}_{Y,V}\left[\left(c - \langle p - e_Y, p' - e_V \rangle\right)^2\right] \\
&= 2\mathbb{E}_{Y,V}\left[\left(c - \langle p - e_Y, p' - e_V \rangle\right)\right] \\
&= 2\left(c - \langle p - \mathbb{P}_Y, p' - \mathbb{P}_V \rangle\right),
\end{aligned}
\tag{35}
$$

and $\frac{\partial^2}{\partial^2 c} L_{\mathrm{CE}}\left(c, \mathbb{P}_Y \otimes \mathbb{P}_V; p, p'\right) > 0$.

Based on the assumption that $\exists A \in \mathscr{F}_{f(X)}$ with $\mathbb{P}_{f(X)}(A) > 0$ we have

$$
\begin{aligned}
&\forall p, p' \in A: \quad h\left(p, p'\right) \neq h^*\left(p, p'\right) \\
\iff& \forall p, p' \in A: L_{\mathrm{CE}}\left(h\left(p, p'\right), \mathbb{P}_{Y|f(X)=p} \otimes \mathbb{P}_{Y|f(X)=p'}; p, p'\right) > L_{\mathrm{CE}}\left(h^*\left(p, p'\right), \mathbb{P}_{Y|f(X)=p} \otimes \mathbb{P}_{Y|f(X)=p'}; p, p'\right) \\
\implies& \int_{A \times A} L_{\mathrm{CE}}\left(h\left(p, p'\right), \mathbb{P}_{Y|f(X)=p} \otimes \mathbb{P}_{Y|f(X)=p'}; p, p'\right) \mathrm{d}\left(\mathbb{P}_{f(X)} \otimes \mathbb{P}_{f(X)}\right)\left(p, p'\right) \\
&> \int_{A \times A} L_{\mathrm{CE}}\left(h^*\left(p, p'\right), \mathbb{P}_{Y|f(X)=p} \otimes \mathbb{P}_{Y|f(X)=p'}; p, p'\right) \mathrm{d}\left(\mathbb{P}_{f(X)} \otimes \mathbb{P}_{f(X)}\right)\left(p, p'\right),
\end{aligned}
\tag{36}
$$

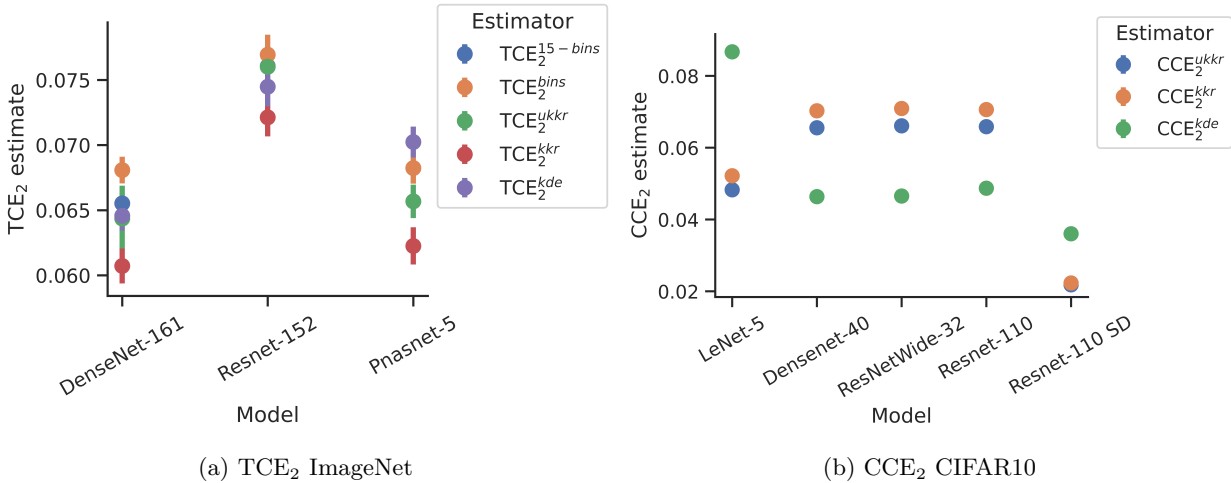

(a) $\mathrm{TCE}_2$ ImageNet

(b) $\mathrm{CCE}_2$ CIFAR10

Figure 5: Different calibration estimates of different models. Most calibration estimates approximately agree with each other. This is in agreement with the similar risk values for each estimator in Table 4 and Table 5.

Table 6: Validation set risk $\sqrt{\hat{\mathscr{R}}_{\mathrm{CE}}} \times 100$ of $\mathrm{TCE}_2$ estimators for VisionTransformer models on various MedMNIST datasets. Lower is better. The error bounds are too wide to make a definitive statement about which estimator is best. Contrary, we can claim that the kernel density estimator performs worst on the Blood and Derma datasets.

| Dataset
Estimator | Blood | Derma | OCT |
|---|---|---|---|
| $\mathrm{TCE}_2^{15-bins}$ | $0.54 \pm 0.12$ | $6.0 \pm 0.42$ | $5.9 \pm 0.94$ |
| $\mathrm{TCE}_2^{bins}$ | $0.52 \pm 0.12$ | $6.0 \pm 0.42$ | $5.87 \pm 0.93$ |
| $\mathrm{TCE}_2^{kde}$ | $0.96 \pm 0.23$ | $7.78 \pm 0.58$ | $5.96 \pm 0.95$ |
| $\mathrm{TCE}_2^{kkr}$ | $0.52 \pm 0.12$ | $5.99 \pm 0.41$ | $5.87 \pm 0.94$ |
| $\mathrm{TCE}_2^{ukkr}$ | $0.52 \pm 0.12$ | $5.99 \pm 0.42$ | $5.87 \pm 0.94$ |

where the inequality follows since $h^*(p, p') = \langle p - \mathbb{P}_{Y|f(X)=p}, p' - \mathbb{P}_{Y|f(X)=p'} \rangle$ is the unique minimizer of $L_{\mathrm{CE}}(., \mathbb{P}_{Y|f(X)=p} \otimes \mathbb{P}_{Y|f(X)=p'}; p, p')$.

From the unique minimizer property also follows that for all $B \in \mathscr{F}_{f(X) \otimes f(X)}$ holds

$$
\begin{aligned}
\int_B & L_{\mathrm{CE}}\left(h(p, p'), \mathbb{P}_{Y|f(X)=p} \otimes \mathbb{P}_{Y|f(X)=p'}; p, p'\right) \mathrm{d}\left(\mathbb{P}_{f(X)} \otimes \mathbb{P}_{f(X)}\right)(p, p') \\
& \geq \int_B L_{\mathrm{CE}}\left(h^*(p, p'), \mathbb{P}_{Y|f(X)=p} \otimes \mathbb{P}_{Y|f(X)=p'}; p, p'\right) \mathrm{d}\left(\mathbb{P}_{f(X)} \otimes \mathbb{P}_{f(X)}\right)(p, p').
\end{aligned}
\tag{37}
$$

Since $\left(\Delta^d \times \Delta^d\right) \setminus (A \times A) \in \mathscr{F}_{f(X) \otimes f(X)}$, it holds

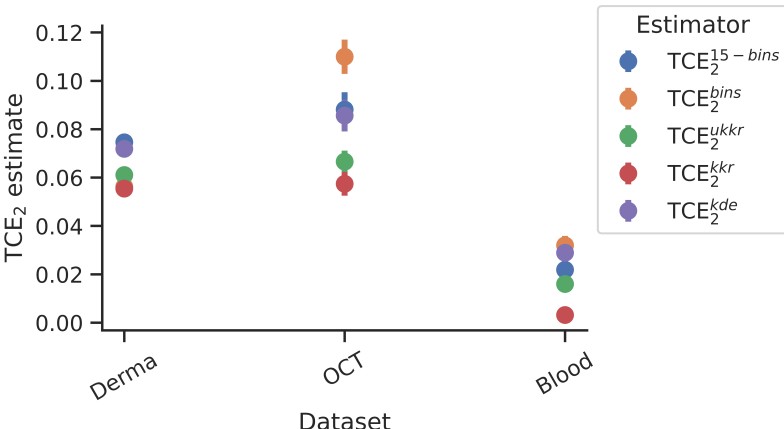

Figure 6: Different top-label confidence calibration estimates of VisionTransformer models trained on MedMNIST datasets. Most calibration estimates approximately agree with each other. Like in previous instances, this demonstrates that the calibration estimate is not directly linked to the respective risk value in Table 6.

$$
\begin{aligned}
&\mathscr{R}_{\mathrm{CE}}\left(h\right) \\
&= \mathbb{E}_{X,X'}\left[L_{\mathrm{CE}}\left(h\left(p,p'\right),\mathbb{P}_{Y|f(X)=p}\otimes\mathbb{P}_{Y|f(X)=p'};p,p'\right)\right] \\
&= \int_{(\Delta^d\times\Delta^d)\backslash(A\times A)} L_{\mathrm{CE}}\left(h\left(p,p'\right),\mathbb{P}_{Y|f(X)=p}\otimes\mathbb{P}_{Y|f(X)=p'};p,p'\right)\mathrm{d}\left(\mathbb{P}_{f(X)}\otimes\mathbb{P}_{f(X)}\right)\left(p,p'\right) \\
&\quad + \int_{A\times A} L_{\mathrm{CE}}\left(h\left(p,p'\right),\mathbb{P}_{Y|f(X)=p}\otimes\mathbb{P}_{Y|f(X)=p'};p,p'\right)\mathrm{d}\left(\mathbb{P}_{f(X)}\otimes\mathbb{P}_{f(X)}\right)\left(p,p'\right) \\
&> \int_{(\Delta^d\times\Delta^d)\backslash(A\times A)} L_{\mathrm{CE}}\left(h^*\left(p,p'\right),\mathbb{P}_{Y|f(X)=p}\otimes\mathbb{P}_{Y|f(X)=p'};p,p'\right)\mathrm{d}\left(\mathbb{P}_{f(X)}\otimes\mathbb{P}_{f(X)}\right)\left(p,p'\right) \\
&\quad + \int_{A\times A} L_{\mathrm{CE}}\left(h^*\left(p,p'\right),\mathbb{P}_{Y|f(X)=p}\otimes\mathbb{P}_{Y|f(X)=p'};p,p'\right)\mathrm{d}\left(\mathbb{P}_{f(X)}\otimes\mathbb{P}_{f(X)}\right)\left(p,p'\right) \\
&= \mathbb{E}_{X,X'}\left[L_{\mathrm{CE}}\left(h^*\left(p,p'\right),\mathbb{P}_{Y|f(X)=p}\otimes\mathbb{P}_{Y|f(X)=p'};p,p'\right)\right] \\
&= \mathscr{R}_{\mathrm{CE}}\left(h^*\right).
\end{aligned}
\tag{38}
$$

## C.2 Proof for Theorem 2

A sketch of the necessity of Theorem 2 is given in Figure 7, which also illustrates the proof.

*Proof.* We use $P \coloneqq f\left(X\right)$ and $f\left(p,p'\right) \coloneqq \begin{cases}\left(h\left(p,p'\right)-h^*\left(p,p'\right)\right)^2, & p,p' \in \mathscr{P}_Y \\ 0, & \text{else,}\end{cases}$ for simplicity. It is continuous at every point in which $h$ and $h^*$ are continuous. We denote with $D$ the $\mathbb{P}_P$-null set of $p$'s for which $h$ and $h^*$

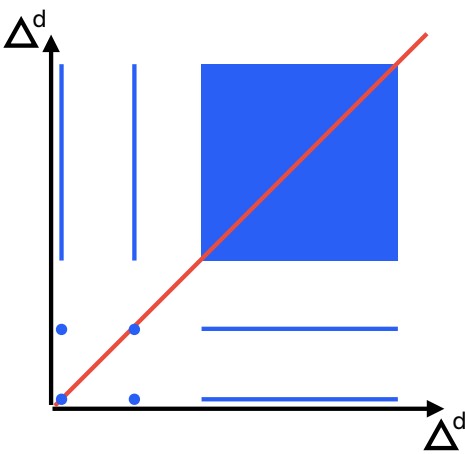

Figure 7: Blue indicates a possible support set $\mathrm{supp}\left(\mathbb{P}_P \otimes \mathbb{P}_P\right) \subseteq \Delta^d \times \Delta^d$, and red line indicates $\left\{(p,p) \mid p \in \Delta^d\right\}$. During training we optimize all blue dots and areas, but during testing we only evaluate their intersection with the red line (a possible null set).

are not continuous at point $(p, p)$. Then,

$$
\begin{aligned}
&\mathbb{E}_P\left[f\left(P, P\right)\right] \\
&= \int_{\mathbb{R}^d} f\left(p, p\right) \mathrm{d}\mathbb{P}_P\left(p\right) \\
&= \int_{\mathrm{supp}(\mathbb{P}_P)} f\left(p, p\right) \mathrm{d}\mathbb{P}_P\left(p\right) \\
&= \int_{\mathrm{int\,supp}(\mathbb{P}_P)} f\left(p, p\right) \mathrm{d}\mathbb{P}_P\left(p\right) + \int_{\mathrm{bd\,supp}(\mathbb{P}_P)} f\left(p, p\right) \mathrm{d}\mathbb{P}_P\left(p\right) \\
&= \int_{\mathrm{int\,supp}(\mathbb{P}_P)\backslash D} f\left(p, p\right) \mathrm{d}\mathbb{P}_P\left(p\right) + \int_{\mathrm{bd\,supp}(\mathbb{P}_P)} f\left(p, p\right) \mathrm{d}\mathbb{P}_P\left(p\right).
\end{aligned}
\tag{39}
$$

The last line holds since the support of a measure is closed, and, consequently, splitting it up into interior and boundary does not add new 'mass' (Teschl, 2014). For the following, denote with $B\left(p, \epsilon\right) :=\left\{x \in \mathbb{R}^d \mid \|x - p\|_2 < \epsilon\right\}$ the open (euclidean) ball with center $p$ and radius $\epsilon$. Further, since $h$ and $h^*$ are by assumption continuous in the points $\{(p, p) \mid p \in \mathscr{P}_Y \backslash D\}$, it follows that $f$ is also continuous at these points, and, consequently, also lower semicontinuous. We first deal with the interior term by using this lower-semicontinuous property giving

$$
\begin{aligned}
&\int_{\mathrm{int\,supp}(\mathbb{P}_P)\backslash D} f\left(p, p\right) \mathrm{d}\mathbb{P}_P\left(p\right) \\
&= \int_{\mathrm{int\,supp}(\mathbb{P}_P)\backslash D} \liminf_{(p_1, p_2) \to (p, p)} f\left(p_1, p_2\right) \mathrm{d}\mathbb{P}_P\left(p\right) \\
&= \lim_{\epsilon \to 0} \int_{\mathrm{int\,supp}(\mathbb{P}_P)\backslash D} \inf\left\{f\left(p_1, p_2\right) \mid (p_1, p_2) \in B\left((p, p), \epsilon\right) \backslash \{(p, p)\}\right\} \mathrm{d}\mathbb{P}_P\left(p\right) \\
&\leq \lim_{\epsilon \to 0} \int_{\mathrm{int\,supp}(\mathbb{P}_P)\backslash D} \mathrm{ess\,inf}\left\{f\left(p_1, p_2\right) \mid (p_1, p_2) \in B\left((p, p), \epsilon\right) \backslash \{(p, p)\}\right\} \mathrm{d}\mathbb{P}_P\left(p\right).
\end{aligned}
\tag{40}
$$

Since $p \in \mathrm{int\,supp}\left(\mathbb{P}_P\right)$, it holds $\mathbb{P}_P\left(B\left(p, \epsilon/2\right) \backslash \{p\}\right) > 0$ for all $\epsilon > 0$ following the definition of int and supp (Teschl, 2014). Let $\mathrm{Sq}\left(p, \epsilon\right) = \left\{x \in \mathbb{R}^d \mid \|p - x\|_\infty < \epsilon\right\}$ be the open hypercube with center $p$ and side length $2\epsilon$.

It holds $B(p, \epsilon) \subset \mathrm{Sq}(p, \epsilon)$ and $\mathrm{Sq}\left(p, \sqrt{d}\epsilon\right) \subset B(p, \epsilon)$. Then

$$
\begin{aligned}
0 &< \mathbb{P}_P\left(B\left(p, \sqrt{2d}\epsilon\right) \setminus p\right) \\
&\leq \mathbb{P}_P\left(\mathrm{Sq}\left(p, \sqrt{2d}\epsilon\right) \setminus \{p\}\right) \\
&= \sqrt{\mathbb{P}_P \otimes \mathbb{P}_P\left(\left(\mathrm{Sq}\left(p, \sqrt{2d}\epsilon\right) \setminus \{p\}\right) \times \left(\mathrm{Sq}\left(p, \sqrt{2d}\epsilon\right) \setminus \{p\}\right)\right)} \\
&\leq \sqrt{\mathbb{P}_P \otimes \mathbb{P}_P\left(\left(\mathrm{Sq}\left(p, \sqrt{2d}\epsilon\right) \times \mathrm{Sq}\left(p, \sqrt{2d}\epsilon\right)\right) \setminus \{(p, p)\}\right)} \\
&= \sqrt{\mathbb{P}_P \otimes \mathbb{P}_P\left(\mathrm{Sq}\left((p, p), \sqrt{2d}\epsilon\right) \setminus \{(p, p)\}\right)} \\
&\leq \sqrt{\mathbb{P}_P \otimes \mathbb{P}_P\left(B\left((p, p), \epsilon\right) \setminus \{(p, p)\}\right)}.
\end{aligned}
\tag{41}
$$

Consequently, $B\left((p, p), \epsilon\right) \setminus \{(p, p)\}$ is not a null set (w.r.t. $\mathbb{P}_P \otimes \mathbb{P}_P$), and, thus,

$$
\begin{aligned}
&\operatorname*{ess\,inf} \{f(p_1, p_2) \mid (p_1, p_2) \in B\left((p, p), \epsilon\right) \setminus \{(p, p)\}\} \\
&\leq \int_{B((p,p),\epsilon)\setminus\{(p,p)\}} f(p_1, p_2)\, \mathrm{d}\left(\mathbb{P}_P \otimes \mathbb{P}_P\right)(p_1, p_2) \\
&\leq \int_{B((p,p),\epsilon)} f(p_1, p_2)\, \mathrm{d}\left(\mathbb{P}_P \otimes \mathbb{P}_P\right)(p_1, p_2) \\
&\leq \mathbb{E}\left[f\left(P, P'\right)\right] = 0,
\end{aligned}
\tag{42}
$$

where we used the given assumption $h\left(P, P'\right) \overset{a.s.}{=} h^*\left(P, P'\right)$ in the last line. Continuing Equation (40) it follows

$$
\lim_{\epsilon \to 0} \int_{\mathrm{int\,supp}(\mathbb{P}_P)\setminus D} \underbrace{\operatorname*{ess\,inf} \{f(p_1, p_2) \mid (p_1, p_2) \in B\left((p, p), \epsilon\right) \setminus \{(p, p)\}\}}_{=0}\, \mathrm{d}\mathbb{P}_P(p) = 0.
\tag{43}
$$

Next, we deal with the boundary of the support. Since we assume that it consists of at most countably infinite elements with probability mass (which we denote as $\{p_1, \dots\}$, it holds

$$
\begin{aligned}
&\int_{\mathrm{bd\,supp}(\mathbb{P}_P)} f(p, p)\, \mathrm{d}\mathbb{P}_P(p) \\
&= \int_{\{p_1, \dots\}} f(p, p)\, \mathrm{d}\mathbb{P}_P(p) \\
&= \sum_{p \in \{p_1, \dots\}} f(p, p)\, \mathbb{P}_P(p) \\
&= \sum_{p \in \{p_1, \dots\}} \sum_{p' \in \{p_1, \dots\}} \mathbf{1}_{p=p'} f(p, p') \sqrt{\mathbb{P}_P(p)}\sqrt{\mathbb{P}_P(p')} \\
&= \int_{\{p_1, \dots\} \times \{p_1, \dots\}} \mathbf{1}_{p=p'} f(p, p')\, \mathrm{d}\left(\sqrt{\mathbb{P}_P} \otimes \sqrt{\mathbb{P}_P}\right)(p, p') \\
&= \int_{\{(p,p) \mid p \in \{p_1, \dots\}\}} f(p, p')\, \mathrm{d}\left(\sqrt{\mathbb{P}_P} \otimes \sqrt{\mathbb{P}_P}\right)(p, p') \\
&\leq \int_{\mathbb{R}^d} f(p, p')\, \mathrm{d}\left(\sqrt{\mathbb{P}_P} \otimes \sqrt{\mathbb{P}_P}\right)(p, p') \\
&= 0,
\end{aligned}
\tag{44}
$$

where the last line holds since for all $A \in \mathscr{F}_{P \times P}$ we have $\sqrt{\mathbb{P}_P} \otimes \sqrt{\mathbb{P}_P}(A) = 0 \iff \mathbb{P}_P \otimes \mathbb{P}_P(A) = 0$.

From Equation (43) and Equation (44) follows that $f\left(P, P'\right) \overset{a.s.}{=} 0 \implies f(P, P) \overset{a.s.}{=} 0$. Consequently, we have $h\left(P, P'\right) \overset{a.s.}{=} h^*\left(P, P'\right) \implies h(P, P) \overset{a.s.}{=} h^*(P, P)$. $\qquad\square$

**Remark.** *Note that any random variable with outcomes restricted to $\Delta^d = \left\{ (p_1, \ldots, p_d)^\intercal \in [0,1]^d \mid \sum_{i=1}^d p_i = 1 \right\} \subset \mathbb{R}^d$ has a singular distribution, since $\Delta^d$ is a null set with respect to the d dimensional Lebesgue measure $\lambda^d$. This would then fall outside of the conditions stated in Theorem 2. However, we can circumvent this simply by transforming (bijectively) the outcome space to $\Delta_r^d = \left\{ (p_1, \ldots, p_{d-1})^\intercal \in [0,1]^{d-1} \mid 0 \le \sum_{i=1}^{d-1} p_i \le 1 \right\} \subset \mathbb{R}^{d-1}$, which has non-zero mass according to the $d-1$ dimensional Lebesgue measure $\lambda^{d-1}$.*

## C.3 Proofs for Section 3.2

We give proofs of various statements of Section 3.2.

### Proof for Equation (27)

Instead of proving the whole kernel ridge regression approach end-to-end, we bring Equation (27) into the form of an ordinary kernel ridge regression objective and then show that our solution in Equation (28) matches the ordinary solution.

For this, define $\tilde{Y}_i \coloneqq \mathrm{vec}_i \left( \boldsymbol{\Delta}_{Yf(X)}^\intercal \boldsymbol{\Delta}_{Yf(X)} \right) = \left\langle f\left(X_{i \bmod n+1}\right) - e_{Y_{i \bmod n+1}}, f\left(X_{\lceil i/n \rceil}\right) - e_{Y_{\lceil i/n \rceil}} \right\rangle \in \mathbb{R}$ and $\tilde{F}_i \coloneqq \left( f\left(X_{i \bmod n+1}\right), f\left(X_{\lceil i/n \rceil}\right) \right) \in \Delta^d \times \Delta^d$ with $i = 1 \ldots n^2$, and $\tilde{h}(\tilde{p}) \coloneqq h(\tilde{p}_1, \tilde{p}_2)$ for $\tilde{p} \in \Delta^d \times \Delta^d$, as well as $\tilde{\mathscr{H}} \coloneqq \mathscr{H} \otimes \mathscr{H}$ with $\tilde{\phi}(\tilde{p}) \coloneqq (\phi \otimes \phi)(\tilde{p}_1, \tilde{p}_2)$.

Then, we can write Equation (27) as

$$
\begin{aligned}
\hat{\mathscr{R}}_{\mathrm{CE}, \lambda}(g) &= \frac{1}{n^2} \sum_{i=1}^n \sum_{j=1}^n \left( \left\langle f(X_i) - e_{Y_i}, f(X_j) - e_{Y_j} \right\rangle - h\left( f(X_j), f(X_j) \right) \right)^2 + \lambda \|g\|_{\mathscr{H} \otimes \mathscr{H}}^2 \\
&= \frac{1}{n^2} \sum_{i=1}^{n^2} \left( \tilde{Y}_i - \left\langle \tilde{g}, \tilde{\phi}(\tilde{F}_i) \right\rangle_{\tilde{\mathscr{H}}} \right)^2 + \lambda \|\tilde{g}\|_{\tilde{\mathscr{H}}}^2 ,
\end{aligned}
\tag{45}
$$

which is ordinary kernel ridge regression in the last line (Bach, 2024). Bach (2024) shows its unique minimum is reached under certain assumptions if

$$
\tilde{g} = \left( \tilde{Y}_1, \ldots, \tilde{Y}_{n^2} \right) \left( \tilde{K}_{f(X)} + \lambda n^2 I \right)^{-1} \left( \tilde{\phi}(\tilde{F}_1), \ldots, \tilde{\phi}(\tilde{F}_{n^2}) \right)^\intercal
\tag{46}
$$

with $\left[ \tilde{K}_{f(X)} \right]_{ij} = \left\langle \tilde{\phi}(\tilde{F}_i), \tilde{\phi}(\tilde{F}_j) \right\rangle_{\tilde{\mathscr{H}}}$. Now, to reach our solution, note that it holds $\left\langle \tilde{\phi}(\tilde{F}_i), \tilde{\phi}(\tilde{p}) \right\rangle_{\tilde{\mathscr{H}}} = k\left( f\left(X_{i \bmod n+1}\right), \tilde{p}_1 \right) k\left( f\left(X_{\lceil i/n \rceil}\right), \tilde{p}_2 \right)$ and $\tilde{K}_{f(X)} = \mathbf{K}_{f(X)} \otimes \mathbf{K}_{f(X)}$, which gives

$$
\begin{aligned}
&\left\langle \tilde{g}, \tilde{\phi}(\tilde{p}) \right\rangle_{\tilde{\mathscr{H}}} \\
&= \left( \tilde{Y}_1, \ldots, \tilde{Y}_{n^2} \right) \left( \tilde{K}_{f(X)} + \lambda n^2 I \right)^{-1} \left( k\left( f\left(X_{1 \bmod n+1}\right), \tilde{p}_1 \right) k\left( f\left(X_{\lceil 1/n \rceil}\right), \tilde{p}_2 \right), \ldots, k\left( f\left(X_{n^2 \bmod n+1}\right), \tilde{p}_1 \right) k\left( f\left(X_{\lceil n^2/n \rceil}\right), \tilde{p}_2 \right) \right)^\intercal \\
&= \mathrm{vec}\left( \boldsymbol{\Delta}_{Yf(X)}^\intercal \boldsymbol{\Delta}_{Yf(X)} \right) \left( \mathbf{K}_{f(X)} \otimes \mathbf{K}_{f(X)} + \lambda n^2 I \right)^{-1} \left( \mathbf{k}_{f(X)}(\tilde{p}_1) \otimes \mathbf{k}_{f(X)}(\tilde{p}_2) \right)^\intercal .
\end{aligned}
\tag{47}
$$

The last line is the predictor we stated in Equation (28).

### Proof for Equation (29)

By definition of $h_{\mathrm{kkr}}$ and by using the eigenvalue decomposition $\mathbf{K}_{f(X)} = Q_{f(X)} \Lambda_{f(X)} Q_{f(X)}^\intercal$, we have

$$h_{\mathrm{kkr}}\left(p, p'\right)$$
$$:= \mathrm{vec}^{\mathsf{T}}\left(\boldsymbol{\Delta}_{Yf(X)}^{\mathsf{T}}\boldsymbol{\Delta}_{Yf(X)}\right)\left(\mathbf{K}_{f(X)} \otimes \mathbf{K}_{f(X)} + \lambda n^2 I\right)^{-1}\left(\mathbf{k}_{f(X)}\left(p\right) \otimes \mathbf{k}_{f(X)}\left(p'\right)\right)$$
$$= \mathrm{vec}^{\mathsf{T}}\left(\boldsymbol{\Delta}_{Yf(X)}^{\mathsf{T}}\boldsymbol{\Delta}_{Yf(X)}\right)\left(Q_{f(X)} \otimes Q_{f(X)}\right)\left(\Lambda_{f(X)} \otimes \Lambda_{f(X)} + \lambda n^2 I\right)^{-1}\left(Q_{f(X)}^{\mathsf{T}} \otimes Q_{f(X)}^{\mathsf{T}}\right)\left(\mathbf{k}_{f(X)}\left(p\right) \otimes \mathbf{k}_{f(X)}\left(p'\right)\right)$$
$$= \left(\left(Q_{f(X)}^{\mathsf{T}} \otimes Q_{f(X)}^{\mathsf{T}}\right)\mathrm{vec}\left(\boldsymbol{\Delta}_{Yf(X)}^{\mathsf{T}}\boldsymbol{\Delta}_{Yf(X)}\right)\right)^{\mathsf{T}}\left(\Lambda_{f(X)} \otimes \Lambda_{f(X)} + \lambda n^2 I\right)^{-1}\left(Q_{f(X)}^{\mathsf{T}} \otimes Q_{f(X)}^{\mathsf{T}}\right)\left(\mathbf{k}_{f(X)}\left(p\right) \otimes \mathbf{k}_{f(X)}\left(p'\right)\right).$$

(48)

Note it holds that $(A \otimes B)\,\mathrm{vec}\,(C) = \mathrm{vec}\left(BCA^{\mathsf{T}}\right)$ for matrices $A, B, C$ and $\mathrm{vec}^{\mathsf{T}}\,(A)\,\mathrm{vec}\,(B) = \mathrm{tr}\left(A^{\mathsf{T}}B\right)$. Then, with the Hadamard product $\odot$ and $\tilde{\Lambda}_X \in \mathbb{R}^{n \times n}$ with $\left[\tilde{\Lambda}_{f(X)}\right]_{ij} := \frac{1}{(\Lambda_{f(X)})_{ii}(\Lambda_{f(X)})_{jj} + \lambda n^2}$, we have

$$\left(\left(Q_{f(X)}^{\mathsf{T}} \otimes Q_{f(X)}^{\mathsf{T}}\right)\mathrm{vec}\left(\boldsymbol{\Delta}_{Yf(X)}^{\mathsf{T}}\boldsymbol{\Delta}_{Yf(X)}\right)\right)^{\mathsf{T}}\left(\Lambda_{f(X)} \otimes \Lambda_{f(X)} + \lambda n^2 I\right)^{-1}\left(Q_{f(X)}^{\mathsf{T}} \otimes Q_{f(X)}^{\mathsf{T}}\right)\left(\mathbf{k}_{f(X)}\left(p\right) \otimes \mathbf{k}_{f(X)}\left(p'\right)\right)$$
$$= \sum_{i=1}^{n^2} \mathrm{vec}_i\left(Q_{f(X)}^{\mathsf{T}}\boldsymbol{\Delta}_{Yf(X)}^{\mathsf{T}}\boldsymbol{\Delta}_{Yf(X)}Q_{f(X)}\right)\mathrm{vec}_i\left(Q_{f(X)}^{\mathsf{T}}\mathbf{k}_{f(X)}\left(p\right)\mathbf{k}_{f(X)}^{\mathsf{T}}\left(p'\right)Q_{f(X)}\right)\left[\tilde{\Lambda}_{f(X)}\right]_{ij}$$
$$= \mathrm{tr}\left(Q_{f(X)}^{\mathsf{T}}\mathbf{k}_{f(X)}\left(p\right)\mathbf{k}_{f(X)}^{\mathsf{T}}\left(p'\right)Q_{f(X)}\left(\tilde{\Lambda}_{f(X)} \odot Q_{f(X)}^{\mathsf{T}}\boldsymbol{\Delta}_{Yf(X)}^{\mathsf{T}}\boldsymbol{\Delta}_{Yf(X)}Q_{f(X)}\right)\right)$$
$$= \mathbf{k}_{f(X)}^{\mathsf{T}}\left(p'\right)Q_{f(X)}\left(\tilde{\Lambda}_{f(X)} \odot Q_{f(X)}^{\mathsf{T}}\boldsymbol{\Delta}_{Yf(X)}^{\mathsf{T}}\boldsymbol{\Delta}_{Yf(X)}Q_{f(X)}\right)Q_{f(X)}^{\mathsf{T}}\mathbf{k}_{f(X)}\left(p\right),$$

(49)

which shows Equation (29).

**Proof for Equation (30)**

Given the definition of $H^{\mathrm{kkr}}$ we have

$$H_{ij}^{\mathrm{kkr}} = \left[\mathbf{K}_{f(X)f(X')}^{\mathsf{T}}Q_{f(X)}\left(\tilde{\Lambda}_{f(X)} \odot Q_{f(X)}^{\mathsf{T}}\boldsymbol{\Delta}_{Yf(X)}^{\mathsf{T}}\boldsymbol{\Delta}_{Yf(X)}Q_{f(X)}\right)Q_{f(X)}^{\mathsf{T}}\mathbf{K}_{f(X)f(X')} \in \mathbb{R}^{n' \times n'}\right]_{ij}$$
$$= \mathbf{k}_{f(X)}^{\mathsf{T}}\left(f\left(X_i'\right)\right)Q_{f(X)}\left(\tilde{\Lambda}_{f(X)} \odot Q_{f(X)}^{\mathsf{T}}\boldsymbol{\Delta}_{Yf(X)}^{\mathsf{T}}\boldsymbol{\Delta}_{Yf(X)}Q_{f(X)}\right)Q_{f(X)}^{\mathsf{T}}\mathbf{k}_{f(X)}\left(f\left(X_j'\right)\right) \in \mathbb{R}^{n' \times n'},$$

(50)

where the last line follows since $\left[\mathbf{k}_{f(X)}\left(f\left(X_j'\right)\right)\right]_i = k\left(f\left(X_i\right), f\left(X_j'\right)\right) = \left[\mathbf{K}_{f(X)f(X')}\right]_{ij}$.

