# OpenReview forum: "Optimizing Estimators of Squared Calibration Errors in Classification"
_TMLR — Accepted by TMLR_

### Review · Reviewer_5D5w · 2024-11-15

**Summary Of Contributions:**

This paper introduces a novel framework to enhance the calibration of classification models, a topic that has gained increasing interest in recent literature. The paper proposes achieving improved calibration by directly optimizing estimators of squared calibration errors. This approach not only aims to refine the calibration process, but also can offer practical guidance on selecting and tuning these estimators to enhance their effectiveness.

**Audience:**

Yes

**Claims And Evidence:**

Yes

**Requested Changes:**

Please refer to the **Weaknesses** in the section above. Overall, this is a solid contribution to the literature, once the weaknesses mentioned are addressed.

**Strengths And Weaknesses:**

**Strengths**
* The paper introduces a mean-squared-error based risk framework that enables evaluation and optimization of squared calibration error estimators in real-world settings.
* It also leverages the bilinear structure of squared calibration errors and reformulates the calibration task to a regression problem. The paper also introduces kernel ridge regression estimators for the calibration task.
* The paper benchmarks existing calibration estimators against the proposed kernel ridge regression based estimators on standard image classification tasks and illustrates the benefits of the proposed calibration framework in practice.
* The proofs, to the best of my knowledge, seem kosher.

**Weaknesses**

* The presentation although technically precise could benefit from greater accessibility. Currently the paper is tailored to an expert audience well versed into the issues and approaches on calibration of classification models. Including a motivating example to illustrate the nature of the problem and the role of squared calibration errors would be helpful to the readers as in other papers (e.g., Vaicenavicius et al., 2019; Gupta et al., 2022).
* Related to the above, currently the way the pipeline (Section 3.2.2) is presented does not provide sufficient clarity to practitioners in terms of how the proposed strategy can be readily adopted. It would be beneficial for the practitioners if the components can be outlined in an algorithm form, so that input/steps can be followed practically.
* The paper addresses the computational complexity of various estimators and outlines strategies to mitigate these challenges in practical scenarios. However, including runtime comparisons for the real-world settings examined would significantly enhance the discussion. Concrete benchmarks would provide readers with a clearer understanding of the trade-offs between computational efficiency and estimator performance.

---

> ### Author Response · Authors · 2025-01-13
> **Improved accessibility and runtime results**
>
> Thank you for your thorough and constructive review. We address all concerns and questions in the following.
>
> > Q: ``The presentation although technically precise could benefit from greater accessibility. Currently the paper is tailored to an expert audience well versed into the issues and approaches on calibration of classification models. Including a motivating example to illustrate the nature of the problem and the role of squared calibration errors would be helpful to the readers as in other papers (e.g., Vaicenavicius et al., 2019; Gupta et al., 2022).''
>
> A: We agree that the initial writing assumes too much expertise from the reader.
> We add an additional section (now Section 2.1) that illustrates the problem of calibration in a more accessible manner, similar to the other papers.
>
> > Q: ``Related to the above, currently the way the pipeline (Section 3.2.2) is presented does not provide sufficient clarity to practitioners in terms of how the proposed strategy can be readily adopted. It would be beneficial for the practitioners if the components can be outlined in an algorithm form, so that input/steps can be followed practically.''
>
> A: We also include an algorithm in Section 3.2.2 for the Calibration-evaluation pipeline to make our work more approachable for practitioners (c.f. Algorithm 1).
>
> > Q: ``The paper addresses the computational complexity of various estimators and outlines strategies to mitigate these challenges in practical scenarios. However, including runtime comparisons for the real-world settings examined would significantly enhance the discussion. Concrete benchmarks would provide readers with a clearer understanding of the trade-offs between computational efficiency and estimator performance.''
>
> A: We provide the individual runtimes of the estimators used in the CIFAR10 experiments in Figure 3. A respective discussion of these additional results is also added.
>
> Please open the revised PDF to see the various additions (marked in red).

---

### Review · Reviewer_mJox · 2024-11-16

**Summary Of Contributions:**

The authors introduce a novel framework to optimize and compare calibration error estimators for classification tasks
My understanding is that the authors are proposing a richer calibration estimator which is more faithful to the canonical calibration estimator. This is in contrast to other more simpler versions of calibration like top-label (confidence) calibration error, which is the dominant notion of calibration in the literature.

**Audience:**

Yes

**Claims And Evidence:**

Yes

**Requested Changes:**

Minor:
- Page 3, under Sec. 2.3: "degree a classifier" should be "degree to which a classifier..."
- Results in Fig. 2 should have higher dpi.

**Strengths And Weaknesses:**

Strengths:
- Well-written and clear motivation and explanations. Mathematical notation is clear.
- Sec. 2 provides a summarization and framework for calibration metrics and estimators.
- Proposed method is mathematically justified and captures a fuller notion of calibration than metrics used in the literature.

Weaknesses:
- My biggest question is about computational complexity and runtime. As a practitioner who wants to quantify calibration, what is the cost of using this method over other more simpler metrics of calibration?
- While evaluations seem thorough, it would be nice to see experiments with more recent transformer-style architectures.
- Overall, the results in tables and figures are not striking, but for the purposes of a venue like TMLR is ok for me.

---

> ### Author Response · Authors · 2025-01-13
> **Runtime and VisionTransformer Results**
>
> We appreciate the time and effort that went into reviewing our manuscript.
> We adopted the requested minor changes.
> In the following, we clarify your questions and alleviate your concerns.
>
> > Q: ``My biggest question is about computational complexity and runtime. As
> a practitioner who wants to quantify calibration, what is the cost of using
> this method over other more simpler metrics of calibration?''
>
> A: The major novelty of our work is that we introduce a risk-based optimization procedure for common calibration (error) estimators.
> The benefit is that we can compare and rank the correctness of various calibration estimators, including the popular binning-based estimator.
> However, this implies additional computational costs since every calibration estimator (including its hyperparameters) requires an evaluation of its risk.
> We added an additional discussion of the computational costs in Section 3.2.2.
> The added computational costs of our method are similar to the added computational costs of hyperparameter optimization in conventional machine learning (Bischl et al., 2023).
> Additionally, we provide the individual runtimes of the estimators used in the CIFAR10 experiments in Figure 3.
> While the benefits for top-label calibration errors may be minor, our contribution enables a more meaningful evaluation of the more difficult notion of canonical calibration, where the binning-based estimator is not feasible.
>
> > Q: ``While evaluations seem thorough, it would be nice to see experiments with more recent transformer-style architectures.''
>
> A: We evaluate a VisionTransformer classifier (Dosovitskiy et al., 2020) on various MedMNIST datasets (Yang et al., 2021) according to the same experimental procedure described in Section 5.
> The logits are generated based on the code in https://github.com/tcapelle/fastai_timm.
> The results for the calibration estimations are presented in the figure in https://imgur.com/a/6An48oz.
> The respective risk values are shown in the following table.
> Overall, the results are similar to the ones presented in the main paper.
>
>
> | Dataset           | Blood           | Derma           | OCT             |
> |-------------------|-----------------|-----------------|-----------------|
> | TCE$_2^{15-bins}$ | 0.54 $\pm$ 0.12 | 6.0 $\pm$ 0.42  | 5.9 $\pm$ 0.94  |
> | TCE$_2^{bins}$    | 0.52 $\pm$ 0.12 | 6.0 $\pm$ 0.42  | 5.87 $\pm$ 0.93 |
> | TCE$_2^{kde}$     | 0.96 $\pm$ 0.23 | 7.78 $\pm$ 0.58 | 5.96 $\pm$ 0.95 |
> | TCE$_2^{kkr}$     | 0.52 $\pm$ 0.12 | 5.99 $\pm$ 0.41 | 5.87 $\pm$ 0.94 |
> | TCE$_2^{ukkr}$    | 0.52 $\pm$ 0.12 | 5.99 $\pm$ 0.42 | 5.87 $\pm$ 0.94 |
>
>
> **References:**
>
> Bischl, B., Binder, M., Lang, M., Pielok, T., Richter, J., Coors, S., ... \& Lindauer, M. (2023). Hyperparameter optimization: Foundations, algorithms, best practices, and open challenges. Wiley Interdisciplinary Reviews: Data Mining and Knowledge Discovery, 13(2), e1484.
>
> Yang, J., Shi, R., \& Ni, B. (2021). Medmnist classification decathlon: A lightweight automl benchmark for medical image analysis. In 2021 IEEE 18th International Symposium on Biomedical Imaging (ISBI) (pp. 191-195). IEEE.
>
> Dosovitskiy, A., Beyer, L., Kolesnikov, A., Weissenborn, D., Zhai, X., Unterthiner, T., ... \& Houlsby, N. (2020). An Image is Worth 16x16 Words: Transformers for Image Recognition at Scale. In International Conference on Learning Representations.

---

### Review · Reviewer_3Qdv · 2024-12-23

**Summary Of Contributions:**

The paper proposes a mean-squared error based risk to compare calibration estimators. The main results are a collection of theoretical results for the newly proposed risk, which includes properties of the optimal solution to the proposed risk, a practical framework for finding a better calibration estimators using the proposed risk function, a unified view of binning TCE and kernel density CCE as the optimal solutions of the risk function under different conditions, and some new calibration estimators inspired by the close-form solutions of the risk function under certain assumptions. The paper also provide synthetic data and real-world data to show that the proposed risk function can indeed compare the performance of calibrations estimators.

**Audience:**

Yes

**Claims And Evidence:**

Yes

**Requested Changes:**

Nothing major, see weakness.

**Strengths And Weaknesses:**

Overall, I enjoyed reading the paper and found the idea of evaluating calibration estimators interesting, and the proposed solution makes sense.

Weakness:
I do not have any major complain for the work. The only thing that I think could be improved is the flow of the paper, especially for Section 4, where the first part of section 4 is unifying some existing calibration functions as the optimal solutions of the risk, and the second part talks about new calibration functions that come from close form optimal solutions of the risk. The title of Section 4 does not provide a good summarization of the Section, at least for me.

---

> ### Author Response · Authors · 2025-01-13
> **Changes to Section 4**
>
> We thank you for your positive review. We address your feedback in the following.
> We agree that further improvements to the text will benefit the reader.
> Specifically, we changed the titles of Section 4, 4.1, and 4.2 to be more descriptive:
> - Section 4: "Calibration Estimation Functions" -> "Calibration Error Estimators as Calibration Estimation Functions"
> - Section 4.1: "Binning and Kernel Density" -> "Reframing Binning and Kernel Density based Calibration Estimators"
> - Section 4.2: "Kernel Ridge Regression" -> "Novel Calibration Estimators based on Kernel Ridge Regression"
>
> We also added a few more sentences to connect the end of Section 4.1 (about binning and kernel density estimators) with the beginning of Section 4.2 (our novel estimators):
>
> "Further, our calibration evaluation pipeline allows to compare the risk of different calibration estimators. This motivates to introduce novel calibration
> estimators, especially for canonical calibration errors, to have a larger search space to optimize over. In
> the following, we introduce a novel class of calibration estimators based on kernel ridge regression, which
> minimize the empirical risk in Equation (20) plus a regularization term under typical kernel ridge regression
> assumptions."

---

### Author Response · Authors · 2025-01-13
**General Response**

We are thankful to all the reviewers for their time and efforts. The insightful feedback and constructive suggestions further improve our work. We are motivated by your positive feedback regarding our work, which is "technically precise" (5D5w) with "clear motivation and explanations" (mJox).

We are convinced that our approach is of strong appeal to the TMLR community since "the idea of evaluating calibration estimators [is] interesting" (3Qdv) and it "introduces a novel framework" (5D5w).

Based on the reviewers’ questions and concerns, we offer new insights into computational costs and further improve the clarity of our work.
The major changes are
- a **new background section** introducing the motivation behind model calibration (c.f. new Section 2.1; Reviewer 5D5w),
- an **algorithm box** for our proposed calibration evaluation pipeline with runtime complexity (c.f. Algorithm 1; Reviewer 5D5w),
- a **runtime figure** (c.f. new Figure 3; Reviewer 5D5w and mJox), and
- clearer titles and reading flow for **Section 4** (Reviewer 3Qdv).

Further, we answer all remaining questions and concerns individually.
The main paper and its appendix are updated with all changes marked in red.
(The red ink will be changed to black upon acceptance.)

---

### Decision · Action_Editor_BBZi · 2025-02-10

**Recommendation:** Accept as is

**Comment:**

Overall, all minor concerns were addressed by the authors during the discussion period, and all reviewers were unanimous and decisive on accepting the paper to TMLR. Congratulations to the authors on a solid contribution!

**Audience:**

All reviewers agree this paper is of interest to the TMLR community.

**Claims And Evidence:**

All reviewers agree that the claims made in this paper of a framework that enables evaluation and optimization of squared calibration error estimators are supported through synthetic and real-world experiments.